# DEXMOVE:
# LEARNING TACTILE-GUIDED NON-PREHENSILE MANIPULATION WITH DEXTEROUS HANDS

**Pei Lin[*,1,2], Yuzhe Huang[*,3,2], Wanlin Li[*,2], Chenxi Xiao[†,1], Ziyuan Jiao[†,2]**

[1]School of Information Science and Technology, ShanghaiTech University
[2]Beijing Institute for General Artificial Intelligence
[3]Beihang University
{linpei2024, xiaochx}@shanghaitech.edu.cn
jiaoziyuan@bigai.ai

## ABSTRACT

Non-prehensile manipulation offers a robust alternative to traditional pick-and-place methods for object repositioning. However, learning such skills with dexterous, multi-fingered hands remains largely unexplored, leaving their potential for stable and efficient manipulation underutilized. Progress has been limited by the lack of large-scale, contact-aware non-prehensile datasets for dexterous hands and the absence of wrist–finger control policies. To bridge these gaps, we present DexMove, a tactile-guided non-prehensile manipulation framework for dexterous hands. DexMove combines a scalable simulation pipeline that generates physically plausible wrist–finger trajectories with a wearable device, which captures multi-finger contact data from human demonstrations using vision-based tactile sensors. Using these data, we train a flow-based policy that enables real-time, synergistic wrist–finger control for robust non-prehensile manipulation of diverse tabletop objects. In real-world experiments, DexMove successfully manipulated six objects of varying shapes and materials, achieving a 77.8% success rate. Our method outperforms ablated baselines by 36.6% and improves efficiency by nearly 300%. Furthermore, the learned policy generalizes to language-conditioned, long-horizon tasks such as object sorting and desktop tidying. Project page: https://peilin-666.github.io/projects/DexMove/

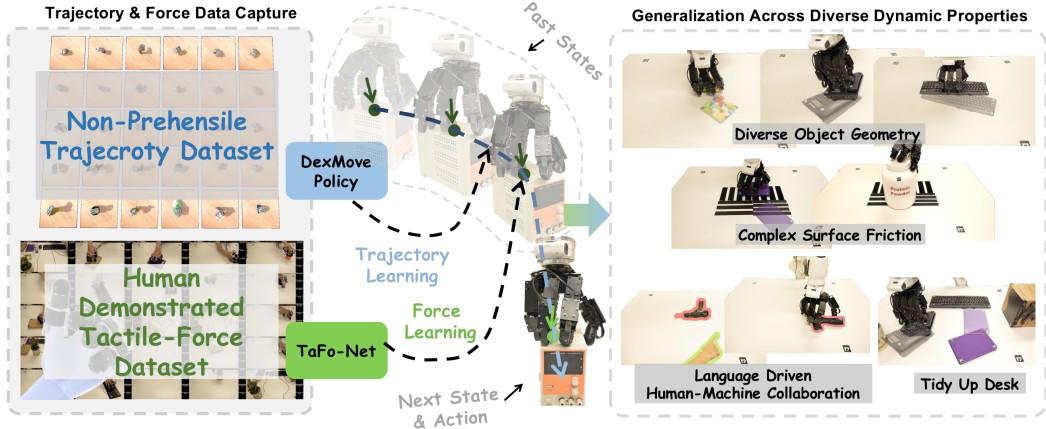

Figure 1: **Overview of DexMove.** The framework integrates synthetic non-prehensile manipulation trajectories and human-demonstrated tactile data to train a flow-matching policy for dexterous hands. The learned policy generalizes across diverse objects, surface frictions, and various language-conditioned tasks such as tidying.

## 1 INTRODUCTION

Non-prehensile manipulation is a fundamental skill that humans routinely employ to interact with their surroundings (Lyu et al., 2025; Cho et al., 2024; Li et al., 2025b). Unlike grasping, it in-

---

* Equal contribution. † Corresponding authors.

duces object motion without lifting the object. This capability is particularly valuable for handling items that are large, heavy, fragile, or geometrically unsuitable for grasping, thereby circumventing the stability and control challenges of full pick-and-place cycles Li et al. (2025a). Consequently, non-prehensile manipulation expands robotic dexterity and applicability of dexterous hands across broader settings, making it a problem worthy of deeper investigation in robotic research.

Given this potential, non-prehensile manipulation has attracted considerable attention in recent years. Most prior work has focused on pushing or pressing objects with robot grippers or rods (Lyu et al., 2025; Cho et al., 2024; Zhou et al., 2023; Suresh et al., 2021). In contrast, the use of dexterous, multi-fingered hands in this context remains largely unexplored (Li et al., 2025b; Wang et al., 2025). Our key insight is that dexterous hands are inherently well-suited for non-prehensile tasks, as they can establish multiple distributed contacts, providing greater stability than a two-finger gripper or pushing rod. This multi-contact capability not only broadens non-prehensile manipulation to thin, cylindrical, and round objects whose dynamics are less predictable during pushing (Lyu et al., 2025; Cho et al., 2024), but also enhances efficiency through coordinated finger motions.

Despite recent advances, deploying dexterous hands for non-prehensile manipulation still faces two key challenges. First, learning generalizable strategies requires large, physically plausible datasets covering variations in object geometry, mass distribution, and surface properties (Lin et al., 2024; Zhu et al., 2024). However, such datasets are not yet available. Acquiring them via teleoperation is limited by inefficiency and compromised force fidelity due to the absence of high-fidelity haptic feedback. Purely simulation-based synthesis is also challenging because of domain gaps, particularly in tactile perception. Second, multi-contact manipulation generates coupled forces and motions across fingers through hand-object dynamics. However, current research lacks a whole-hand motion planner for coordinating multi-contact interactions. Together, the scarcity of scalable, high-fidelity datasets and the absence of force-aware, multi-contact coordination policies hinder progress toward generalizable dexterous non-prehensile manipulation.

To address these challenges, we propose DexMove, a framework for dexterous non-prehensile manipulation (Fig. 1). *First*, to overcome the difficulty of scaling multi-contact interaction data, we build a large-scale simulation pipeline that synthesizes diverse, force-aware wrist–finger trajectories across objects with varying geometry, friction, and mass distribution. *Second*, to exploit tactile information, we develop a wearable system with vision-based tactile sensors that captures fingertip force distributions from human demonstrations. The tactile knowledge is distilled into TaFo-Net, a network that learns a spatiotemporal inter-finger force representation from human demonstrations. *Third*, to seamlessly bridge the trajectory in simulation and tactile data from the real domain, we introduce the DexMove-Policy, a flow-matching network that learns synergistic, tactile-based coordination strategies for jointly controlling the wrist and fingers. **The significance of DexMove lies in three key aspects: (i) the first non-prehensile policy tailored for tactile dexterous hands, (ii) a novel data synthesis paradigm that integrates large-scale trajectory simulation with limited human tactile demonstrations, enabling the incorporation of tactile data with minimal domain gaps. (iii) a novel wearable exoskeleton system with vision-based tactile sensors.**

The remainder of this paper is organized as follows. Sec. 2 reviews prior work on tactile-based non-prehensile manipulation. Sec. 3 details our hybrid data acquisition pipeline, which integrates simulation-based trajectory generation with tactile force data from human demonstrations. Sec. 4 presents the proposed learning framework, including the contact-establishment policy, DexMove-Policy for jointly controlling the wrist and fingers to reposition the object, and tactile-force planning network (TaFo-Net). Sec. 5 reports quantitative and qualitative experiments, including performance evaluation, ablation studies, and downstream applications. Finally, Sec. 6 summarizes our findings, discusses limitations, and outlines directions for future research.

## 2 RELATED WORK

### 2.1 NON-PREHENSILE MANIPULATION

Non-prehensile manipulation involves interacting with objects without grasping (Oller et al., 2024; Yang & Posa, 2024; Lyu et al., 2025), enabling efficiencies beyond repeated pick-and-place cycles. Based on contact dynamics, it is commonly divided into single-contact and multi-contact strategies. Single-contact methods use a tool or end-effector to drive object motion; planar pushing (Mason, 1986; Stüber et al., 2020) is a canonical example where a rod translates or rotates an object. Later work (Chi et al., 2024; Ferrandis et al., 2024) extended this setting to controlled contact breaking

and reestablishment, enabling intermittent push sequences. However, single-contact manipulation is intrinsically less certain because lateral forces are weakly constrained, especially under non-uniform friction or uneven surfaces. Multi-contact manipulation instead distributes forces across multiple contact points, leading to more stable interactions. Prior work (Bhat et al., 2023) showed that even two-point contact can substantially improve stability over single-point strategies. Building on this, we leverage the multiple fingers of a dexterous hand and integrate tactile sensing for closed-loop finger control, improving robustness and efficiency. Moreover, whereas prior approaches often focus on ad hoc object sets (Li et al., 2025b; Wang et al., 2025), our method generalizes non-prehensile manipulation to unseen objects.

## 2.2 Data Collection for Tactile-guided Manipulation

Tactile sensing has been central to robotic manipulation (She et al., 2021; Do et al., 2023; Jin et al., 2023; Qi et al., 2023; Huang et al., 2024; Zhu et al., 2025; Taylor et al., 2024; Higuera et al., 2025). To exploit tactile feedback, recent work increasingly relies on data-driven policy learning (Dong et al., 2021; Suresh et al., 2024; Sun et al., 2025), but scaling tactile data collection remains a key bottleneck. Although simulation can generate data, it is computationally costly due to soft-body dynamics and often suffers from sim-to-real gaps (Si & Yuan, 2022).

Teleoperation offers a domain-gap-free alternative, yet most systems focus on parallel-jaw grippers (Wu et al., 2025a). Dexterous multi-fingered hands are substantially harder to operate without haptic feedback, causing greater tactile-signal variability and lower success rates (Zhang et al., 2025). To address this, exoskeleton-based teleoperation has been explored (Yang, 2025; Zhang et al., 2025; Fang et al., 2025), providing real-time haptics but requiring labor-intensive setups and restricting natural motion, which limits scalability. Another direction is collecting tactile data directly from humans: tactile gloves (Liu et al., 2024; Jiang et al., 2024; Xing et al., 2025) sense distributed hand pressure, but mismatched sensor layouts versus robot hands introduce domain gaps. More recently, isomorphic sensor designs (Zhu et al., 2025; Wu et al., 2025b) enable portable, domain-gap-free tactile capture while preserving natural hand motion.

Motivated by these developments, we propose a hybrid acquisition paradigm that combines simulation scale with real-world fidelity. We generate large-scale motion data in simulation and augment it with real tactile measurements to compensate for limited simulated tactile realism. On the hardware side, an exoskeleton interface transfers human interactions to robotic hands equipped with isomorphic sensors, enabling policies to benefit from both data sources in a complementary way.

## 3 Data Acquisition for Non-prehensile Manipulation

This section introduces a hybrid data synthesis pipeline that incorporates tactile information into the manipulation process. In Sec. 3.1, we present an optimization procedure that generates multi-finger motion trajectories across diverse grasp poses and force levels to guide objects toward target poses. In Sec. 3.2, we complement this with human demonstrations to capture how contact forces are modulated during manipulation. Both datasets support policy training in Sec. 4.

### 3.1 Trajectory Synthesis

#### 3.1.1 Hand-Object Contact Establishment

Non-prehensile manipulation begins by establishing an initial hand–object contact. To generate diverse contact poses, we uniformly sample candidate wrist poses, each defined by rotation $\mathbf{R}_0^{\text{wrist}} \in \mathbb{R}^{3\times3}$ and translation $\mathbf{T}_0^{\text{wrist}} \in \mathbb{R}^3$ following Yang et al. (2021) and subscript $_0$ indicates the initial frame of the entire manipulation sequence. Then, a displacement vector $\mathbf{d}$ from each fingertip to its nearest surface point is computed, and the fingertip is translated along $\mathbf{d}$ until contact. To promote diversity, Gaussian noise $\varepsilon$ perturbs the direction, yielding $\hat{\mathbf{d}} = \mathbf{d} + \varepsilon$. Given the fingertips' positions $\mathbf{P}_0^{\text{TIP}} \in \mathbb{R}^{3\times4}$, the finger joints' angles $\hat{\mathbf{A}}_0^{\text{hand}} \in \mathbb{R}^J$ ( $J$ denotes the number of joints) is synthesized by solving:

$$\hat{\mathbf{A}}_0^{\text{hand}} = \arg\min_{\mathbf{A}_0^{\text{hand}}} \|\text{FK}(\mathbf{A}_0^{\text{hand}}, \mathbf{R}_0^{\text{wrist}}, \mathbf{T}_0^{\text{wrist}}) - \mathbf{P}_0^{\text{TIP}}\|_2 + w_{\text{pinch}} L_{\text{region}}, \qquad (1)$$

$$L_{\text{region}} = \|\mathbf{d}^{\text{TIP}} - \hat{\mathbf{d}}\|_2, \text{ where } \mathbf{d}^{\text{TIP}} = (\mathbf{P}_0^{\text{TIP}} - \mathbf{P}_0^{\text{DIP}}) \qquad (2)$$

where FK is the forward kinematic function, which can output positions of fingertips, and $L_{\text{region}}$ encourages contacts within the tactile sensor's effective region. $\mathbf{d}^{\text{TIP}}$ denotes the orientation vector of fingertip from the DIP joint position $\mathbf{P}_0^{\text{DIP}}$ to the fingertip joint position $\mathbf{P}_0^{\text{TIP}}$.

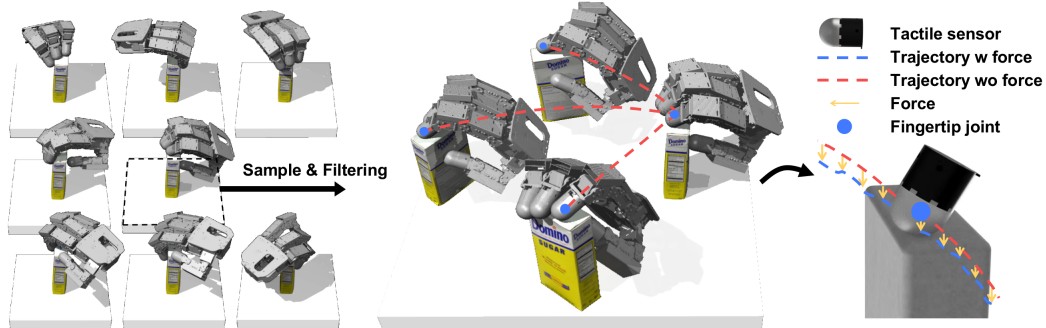

**(a) Generate Grasping Pose**    **(b) Non-prehensile Trajectory**    **(c) Trajectory with Force**

Figure 2: Trajectory data verification and force augmentation. (a,b) After generating initial grasp poses, the set of target directions along which the object can be manipulated is pruned using physics simulation, and the fingertips' trajectories are computed by sampling target object poses. (c) The contact force is synthesized as a surrogate for the finger's penetration depth into the object, and we augment the trajectories with diverse forces.

To build a diverse dataset of hand–object contacts, we use 88 objects from the YCB dataset (Calli et al., 2015), which are then randomly scaled and rotated to produce 352 object layout instances. For each instance, we generate 1,024–2,048 candidate poses. After discarding poses with self-collisions or penetrations, a total of 412k valid configurations remain.

### 3.1.2 SYNTHESIZE FORCE-CONDITIONED TRAJECTORIES

After establishing initial contacts, the object is manipulated toward a target pose through coordinated wrist and finger motions which are parameterized by $\mathbf{A}^{\text{hand}}$, $\mathbf{R}^{\text{wrist}}$, and $\mathbf{T}^{\text{wrist}}$. These motions induce object movement with 3 DoFs: $x/y$ axis, and yaw rotation. While such variables could be optimized using shooting methods (e.g., iLQR (Li & Todorov, 2004)), these approaches are computationally expensive and often yield uncertain trajectories due to unmodeled physical effects. To address these challenges, we employ a rejection-sampling strategy for trajectory synthesis.

We simulate the repositioning process in MuJoCo (Todorov et al., 2012) by incrementally translating the hand along random directions. If all fingertips maintain stable contact after a displacement of 50 cm, the direction is accepted as feasible. From the resulting admissible set, we generate data by uniformly sampling a target object position $\mathbf{P}^{\text{obj}}_{\text{target}} \in \mathbb{R}^3$ and yaw $\omega^{\text{obj}}_{\text{target}} \in \mathbb{R}$ as the target pose.

Under the non-slip assumption, each fingertip's trajectory is computed by rigidly transforming its initial relative contact offset with respect to the object reference point:

$$\mathbf{P}^{\text{tip}}_t = \mathbf{P}^{\text{obj}}_t + \mathbf{R}_z(\omega^{\text{obj}}_t)\big(\mathbf{P}^{\text{TIP}}_0 - \mathbf{P}^{\text{obj}}_0\big), \quad t = 0, \ldots, T, \tag{3}$$

where T is the length of the manipulation sequence, $\mathbf{P}^{\text{obj}}_0$ denotes the initial position of object, $\omega^{\text{obj}}_t$ denotes the interpolated yaw angle from zero to $\omega^{\text{obj}}_{\text{target}}$ at step t, $\mathbf{R}_z(\omega^{\text{obj}}_t)$ is the rotation matrix representing the object's yaw at step $t$, and $\mathbf{P}^{\text{obj}}_t$ denotes the interpolated object translation from the initial origin to the target pose at step $t$.

To model the contact forces along these trajectories, we approximate the fingertip–object normal force $G$ using the indentation depth measured by the vision-based tactile sensor:

$$G \approx D^{\text{sensor}} = r - \text{distance}(\mathbf{P}^{\text{TIP}}_t, \text{surface}), \tag{4}$$

where $r$ is the fingertip radius and $\text{distance}(\cdot)$ denotes the Euclidean distance between the fingertip joint position and the contact object surface. $D^{\text{sensor}}$ is the indentation depth of tactile sensor. As shown in Fig. 2, we further sample multiple force profiles by displacing each fingertip along its contact normal $\vec{\mathbf{n}}$:

$$\hat{\mathbf{P}}^{\text{TIP}}_t = \mathbf{P}^{\text{TIP}}_t + \vec{\mathbf{n}} \cdot \mathcal{N}(0, \sigma), \tag{5}$$

which produces augmented fingertip trajectories with varying force magnitudes. The contact normal $\vec{\mathbf{n}}$ is computed as the direction vector from the contact point to the fingertip $\mathbf{P}^{\text{TIP}}_t$. Given these updated fingertip positions, the corresponding joint and wrist configurations are recovered by solving an inverse kinematics (IK) problem with additional wrist-motion regularization.

Given these force-augmented fingertip trajectories, the joint and wrist configurations are recovered by solving an optimization-based inverse kinematics problem:

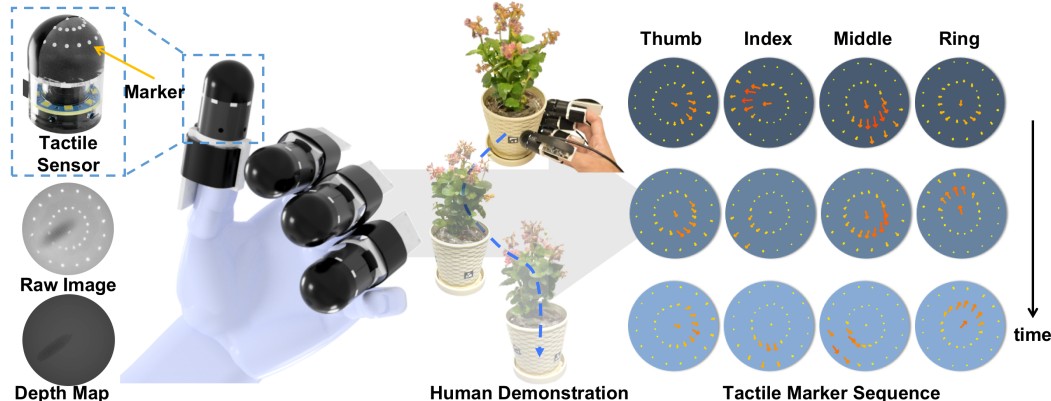

Figure 3: A wearable device for collecting real tactile-force data. The collected force during manipulation is characterized by the displacement of markers within the vision-based tactile sensor.

$$\mathbf{X}^{\text{hand}} = (\mathbf{A}^{\text{hand}}, \mathbf{R}^{\text{wrist}}, \mathbf{T}^{\text{wrist}})_{0:T} \tag{6}$$

$$\hat{\mathbf{X}}^{\text{hand}} = \text{argmin}_{\mathbf{X}^{\text{hand}}} ||\text{FK}(\mathbf{X}^{\text{hand}}) - \hat{\mathbf{P}}^{\text{TIP}}_{0:T}||_2 + w^{\text{wrist}} L^{\text{wrist}}, \tag{7}$$

$$L^{\text{wrist}} = \text{MSE}_{0:T}(\mathbf{R}^{\text{wrist}}, \hat{\mathbf{R}}^{\text{wrist}}) + \text{MSE}_{0:T}(\mathbf{L}^{\text{wrist}}, \hat{\mathbf{L}}^{\text{wrist}}), \tag{8}$$

where MSE denotes the mean squared error. To bias the solution toward finger-driven manipulation, we add $L^{\text{wrist}}$, a penalty on wrist motion that reduces the likelihood of moving the arm outside its workspace. Finally, a filtering step further constrains the robot arm: in simulation, any trajectory that leaves the feasible workspace is discarded. We totally collected 2M sequences.

## 3.2 Tactile Force From Human Demonstrations

The contact forces exerted by individual fingers are important in contact-rich tasks but challenging to estimate in simulation due to two main challenges. First, high-fidelity object dynamics are difficult to model accurately. Second, our rigid-body simulation framework is unable to generate tactile outputs, as it lacks realistic soft-body contact modeling. To address these limitations, we adopt a demonstration-based strategy that infers contact force from historical tactile observations. We developed a wearable device with tactile sensors mounted on human fingers (Fig. 3). The vision-based tactile sensor used here is an R-Tac sensor (Lin et al., 2025), augmented with additional visual markers that enable direct representation of interaction forces. The exoskeleton design allows easy mounting on human fingertips for data collection and subsequent attachment to the robot hand. Its isomorphic design to the robot hand helps minimize the domain gap caused by hardware differences, following the intuition in Fang et al. (2025). Further hardware details are provided in Appx. A.1.

During each trial, we recorded the tactile information as well as the object's target pose and real-time pose throughout the manipulation. The tactile data include the normal force magnitude $G$, estimated from the inferred indentation depth, and tangential (shear) forces derived from 2-D displacement of surface markers (represented as a normalized direction together with a scalar magnitude). This results in a tactile vector field $\mathbf{V} \in \mathbb{R}^{v \times 4}$, where $v = 33$ is the number of markers (Fig. 3). Data were captured at 30 FPS, yielding approximately 300k frames across 20 objects.

## 4 Policies for Non-prehensile Dexterous Manipulation

Next, we present our method for integrating kinematic trajectories with contact force planning learned from human demonstrations. Specifically, we introduce three components of our framework: a policy for establishing contact, the DexMove policy for jointly controlling the wrist and fingers to manipulate objects, and TaFo-Net for coordinating finger forces. Each component is described in detail below.

### 4.1 Establish Contact

A Flowing Matching (FM) (Lipman et al., 2023; Liu et al., 2022) policy is utilized to infer an initial contact hand pose with the object (Fig. 4 (left)). Compared to diffusion policy (Chi et al., 2024), we found FM offers faster training and inference speed. FM aims to generate the hand state

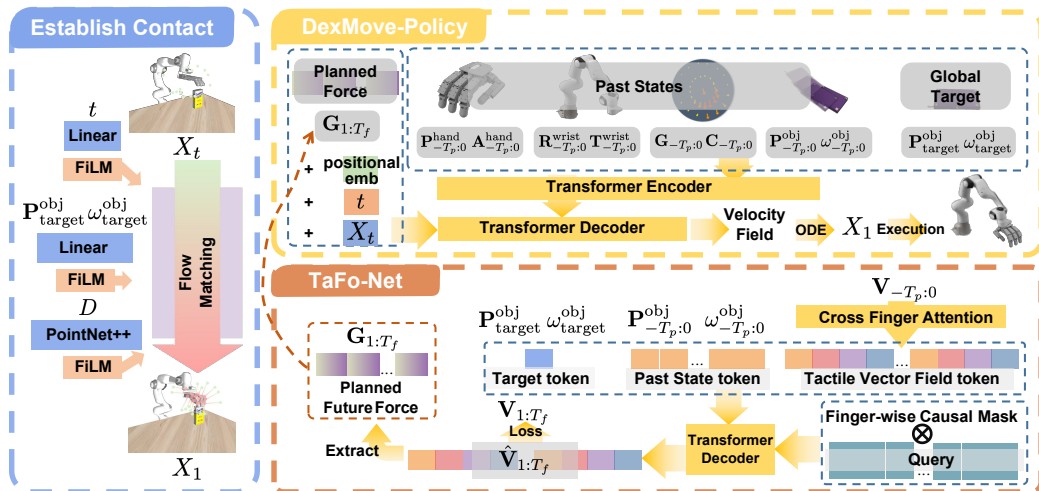

Figure 4: Pipeline for Manipulation. **Establish Contact**: a flow-matching model to predict contact hand pose by fusing the conditions. **DexMove-Policy**: a transformer-based flow-matching network predicts future hand trajectory and states conditioned on force, past states, and global target. **TaFo-Net**: a transformer decoder predicts tactile vector fields (i.e., forces) for DexMove-Policy.

$(\mathbf{A}_0^{\text{hand}}, \mathbf{R}_0^{\text{wrist}}, \mathbf{T}_0^{\text{wrist}})$ conditioned on the object point cloud $D$ and target object pose $(\mathbf{P}_{\text{target}}^{\text{obj}}, \omega_{\text{target}}^{\text{obj}})$. Using the target object pose as a conditioning input, its goal is to generate hand poses capable of moving the object to the desired target. The FM policy learns a time-dependent velocity field $u(\cdot)$ from intermediate samples:

$$X_t = (1-t)X_0 + tX_1,$$

where $t \sim U[0,1]$, $X_0 \sim \mathcal{N}(0, I)$, and $X_1$ is drawn from the distribution of ground-truth samples. The objective is:

$$L_{\text{contact}} = \mathbb{E}\big\|(X_1 - X_0) - u(X_t, t, \text{condition})\big\|^2,$$

so that the velocity field learns to generate feasible contact configurations by integrating the ODE from a randomly sampled $X_0$ to $X_1$. To allow for the acquisition of conditioned features, PointNet++ (Qi et al., 2017) is used to extract features from the point cloud. These conditional features are incorporated into the latent representation of $X_t$ via FiLM (Perez et al., 2017). These details are illustrated in Fig. 4 (left) and further discussed in Appx. A.3.2.

## 4.2 TRAJECTORY LEARNING: DEXMOVE-POLICY

After establishing initial contact, the repositioning process is controlled by DexMove-Policy, a goal-conditioned flow-matching model. Conditioned on the observed state history, the target object pose, and a desired force schedule, the model generates a rollout of future hand states for execution. Specifically, the system state history is represented as:

$$\big\{\mathbf{P}^{\text{hand}}, \mathbf{A}^{\text{hand}}, \mathbf{R}^{\text{wrist}}, \mathbf{T}^{\text{wrist}}, \mathbf{P}^{\text{obj}}, \omega^{\text{obj}}, \mathbf{C}, \mathbf{G}\big\}_{-T_p:0}, \tag{9}$$

where $\mathbf{P}^{\text{hand}} \in \mathbb{R}^{J \times 3}$ denotes joint positions, $\mathbf{C} \in \mathbb{R}^{F \times 3}$ denotes contact positions (each expressed in the fingertip's local frame), and $\mathbf{G} \in \mathbb{R}^F$ denotes the pressing force of each finger, inferred from sensor indentation during execution. Here, $F$ is the number of fingers and $-T_p:0$ is the past $T_p$ steps until now. In addition to historical observations, the network also receives desired finger forces from Sec. 4.3, denoted as $\mathbf{G}_{1:T_f} \in \mathbb{R}^{T_f \times F}$, where $T_f$ is the number of future frames to predict.

The network structure is illustrated in Fig. 4 (top-right). Historical states and the target object pose are fused via cross-attention. The fused data and force commands are passed to a Transformer decoder that predicts the flow-matching velocity field. The predictive sample $X_1$ consists of $T_f$ frames of future hand states:

$$X_1 = \big\{\mathbf{P}^{\text{hand}}, \mathbf{A}^{\text{hand}}, \mathbf{R}^{\text{wrist}}, \mathbf{T}^{\text{wrist}}\big\}_{1:T_f}. \tag{10}$$

### 4.3 Force Planning: TaFo-Net

The trajectories discussed above are conditioned on the desired finger forces $\mathbf{G}_{1:T_f} \in \mathbb{R}^{T_f \times F}$. In this section, we introduce TaFo-Net (Fig. 4 (bottom-right)), a network designed to predict desired forces.

Given the target object pose $(\mathbf{P}_{\text{target}}^{\text{obj}}, \omega_{\text{target}}^{\text{obj}})$, the past $T_p$ frames of object states $(\mathbf{P}_{-T_p:0}^{\text{obj}}, \boldsymbol{\omega}_{-T_p:0}^{\text{obj}})$, and per-finger tactile vector fields $\mathbf{V}_{-T_p:0} \in \mathbb{R}^{T_p \times F \times v \times C}$, TaFo-Net predicts the tactile vector fields for the next $T_f$ frames, $\mathbf{V}_{1:T_f} \in \mathbb{R}^{T_f \times F \times v \times C}$. Here, $v$ is the number of markers per finger, $F$ is the number of fingers and $C$ is the number of channels per marker. The predicted tactile vector fields are used to extract the per-finger pressing forces $\mathbf{G}_{1:T_f}$, which serve as targets for the trajectory policy described earlier. Our key insight is that historical tactile vector fields encode environmental properties implicitly, such as surface friction and contact state. The history of object poses together with the target pose provides a distance-to-go signal, from which future actions can be inferred.

To capture spatial-temporal inter-finger interactions, TaFo-Net has three stages:

(i) **Per-finger spatial encoding.** For each time $t$ and finger $f$, the tactile vector field $\mathbf{V}_{t,f} \in \mathbb{R}^{v \times C}$ is encoded into a finger token $\mathbf{U}_{t,f}$ via a lightweight transformer, enriched with learnable and geometry-informed marker positional embeddings.

(ii) **Cross-finger attention.** At each frame $i$, we form the set $\{\mathbf{U}_{i,1}, \ldots, \mathbf{U}_{i,F}\}$ and apply multi-head self-attention across fingers (CF), augmented with finger-type embeddings $\{\mathbf{g}_f \in \mathbb{R}^D\}$. This produces cross-finger enhanced tokens $\tilde{\mathbf{U}}_{t,f} \in \mathbb{R}^D$, denoted as: $\tilde{\mathbf{U}}_{i,1:F} = \text{CF}(\mathbf{U}_{i,1:F} + \mathbf{g}_{1:F})$.

(iii) **Finger-wise causal temporal attention.** After merging the tactile vector fields with the target and past states, we apply a finger-wise causal mask to the queries so that a query at $i$ can attend only to tokens from times $\leq i$ (across all fingers), preventing information leakage from the future and enabling goal-conditioned, temporally consistent, and cross-finger consistent reasoning.

To enhance robustness, we randomly drop out time steps, fingers, and tactile markers during training. Finally, the model is trained by minimizing a reconstruction loss:

$$\mathcal{L}_{\text{rec}} = \sum_{t=1}^{T_f} \sum_{f=1}^{F} \left\| \widehat{\mathbf{V}}_{t,f} - \mathbf{V}_{t,f} \right\|^2. \tag{11}$$

## 5 Experiment

In this section, we present comprehensive experiments to evaluate the proposed DexMove pipeline. First, we compare our system with other non-prehensile manipulation methods in terms of task performance. Next, we conduct ablation studies to examine the contribution of each technical module as well as system robustness. Finally, we demonstrate downstream applications enabled by our approach. Our hardware settings and implementation details can be found at Appx. A.3.

### 5.1 Performance Benchmark

To the best of our knowledge, non-prehensile manipulation with a tactile dexterous hand remains under-explored, and no publicly available baselines exist. We therefore benchmark our approach against the following baselines: (i) an open-loop replay policy, and (ii) two gripper-based learning methods, CORN from Cho et al. (2024) and DyWA from Lyu et al. (2025). All evaluations are conducted on a desktop under two friction conditions: a clean tabletop and with tape strips. We benchmarked six objects: a randomly assembled LEGO (Fig. 5 (a)), a keyboard, a mouse, a book, a large cylindrical can, and a small cylindrical can. The selected objects span diverse masses, sizes, and geometries, as shown in Fig. 12. Other implementation details are illustrated in Appx. A.5.

We evaluate all baselines using the following metrics: (i) **Success rate**: a trial is successful if the terminal state relative to the target pose is within 10% error in both yaw angle and position, and no accidental self-collision occurs; (ii) **Efficiency**: the time taken to reach the terminal state. Each object is tested in 30 trials, and each trial has identical initial and target poses for different methods.

#### 5.1.1 Comparison on Success Rate

The benchmarked success rates are reported in Tab. 1. First, the Open-Loop baseline simply replays a previously successful DexMove trajectory, yielding the lowest success rate among all policies; without feedback, it cannot handle movement errors that arise during manipulation. Second, the DyWA and CORN baselines, which employ grippers for discrete-contact non-prehensile manipula-

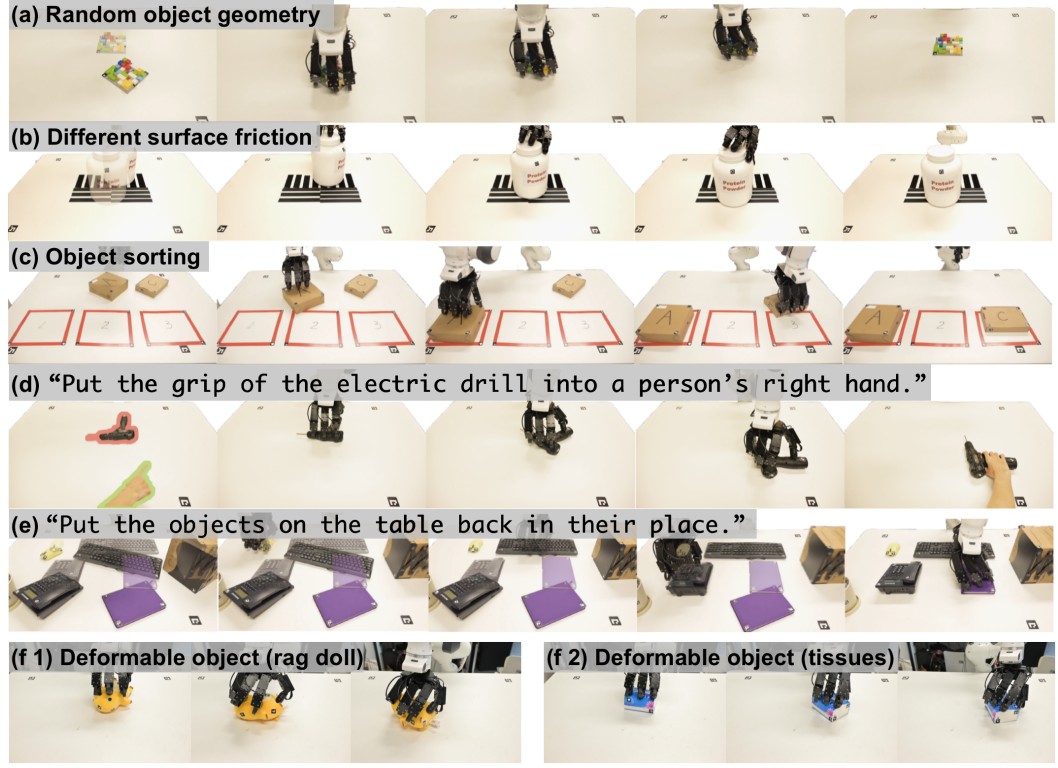

Figure 5: Demonstrations of non-prehensile manipulation. Our learned policy robustly adapts to (a) objects with challenging geometries and (b) varying surface friction. In addition, the learned skill supports reasoning tasks such as (c) object sorting, and language-guided repositioning tasks, including (d) non-prehensile handover and (e) tidying up objects on a desktop, (f) deformable objects (rag dolls and a packet of tissues).

tion, also achieve lower success rates than our policy. Both DyWA and CORN predominantly fail by not achieving the desired rotations, particularly for cylindrical objects. We believe this is due to dependence on a single contact point for repositioning. In contrast, the dexterous hand establishes stable, continuous, multi-surface contacts, enabling accurate rotation of everyday objects.

Furthermore, DexMove demonstrates robustness against non-uniform surface friction (Fric. A vs. Fric. B, where Fric. B is unseen during data collection (Fig. 5 (b))). The performance gap between the two surface conditions is only marginal. By contrast, DyWA and CORN exhibit pronounced performance degradation, reflecting their sensitivity to unmodeled spatial friction variability.

Table 1: Success rate (%) of DexMove under different initial yaw angle errors $\omega_{\text{target}}^{\text{obj}}$ (degrees).

| Method | $0 < \omega_{\text{target}}^{\text{obj}} < 30$ | | $30 < \omega_{\text{target}}^{\text{obj}} < 60$ | | $60 < \omega_{\text{target}}^{\text{obj}} < 90$ | |
|---|---|---|---|---|---|---|
| | Fric. A | Fric. B | Fric. A | Fric. B | Fric. A | Fric. B |
| Open-loop | 36.7 | 10.0 | 23.3 | 0.0 | 3.3 | 0.0 |
| DyWA (Lyu et al., 2025) | 50.0 | 36.7 | 46.7 | 30.0 | 50.0 | 33.3 |
| CORN (Cho et al., 2024) | 43.3 | 36.7 | 46.7 | 40.0 | 43.3 | 43.3 |
| DexMove (Ours) | **86.7** | **86.7** | **80.0** | **83.3** | **70.0** | **60.0** |

### 5.1.2 COMPARISON ON EXECUTION EFFICIENCY

In addition to the success rate, we evaluate efficiency using the average completion time. Timing is measured from the moment the hand or gripper first contacts the object until the object reaches the target pose within the defined success threshold. The results are reported in Tab. 2. Among all comparison groups, DexMove achieves an average completion time less than half that of DyWA and CORN, owing to its use of multi-finger contact and the reduced number of action primitives to reach the target pose. These findings highlight DexMove as a highly efficient manipulation policy.

Table 2: Average execution time (s) across different pushing distances, highlighting the efficiency of DexMove.

| Method | $0 < P < 15$ cm | $15 < P < 30$ cm | $30 < P < 45$ cm |
|---|---|---|---|
| DyWA (Lyu et al., 2025) | 36.1 | 52.2 | 60.6 |
| CORN (Cho et al., 2024) | 41.4 | 54.5 | 62.1 |
| DexMove (Ours) | **8.3** | **10.9** | **12.4** |

## 5.2 ABLATION STUDY

We perform ablation studies to quantify the contribution of each module. The environmental setting is the same as Sec. 5.1. The baselines we compared include: (1) Wrist-Only: using the robot wrist to automatically move objects via a policy, with all finger joints locked after initial contact. (2) Wrist-only* denotes the same setting, but with the wrist controlled through teleoperation. (3) w/o Cross-Finger: the cross-finger attention blocks in TaFo-Net are removed. (4) w/o Shear Force: the tactile vector field excludes shear components, retaining only the normal component. (5) w Heuristic Force: we disable TaFo-Net and replace it with a hand-crafted strategy (if slip is detected, incrementally increase the force by a fixed increment following Lin et al. (2025)). The implementation details are illustrated in Appx. A.5.

From the results in Tab. 3, we observed that DexMove achieved the highest success rate in most cases, underscoring the necessity of using a tactile dexterous hand with active finger control. In the Wrist-Only configuration, flat or planar objects (book and keyboard) can be manipulated with a high success rate. But it rarely succeeds when object shape induces non-coplanar fingertip contacts where finger adjustments become necessary. Without the Cross-Finger module, TaFo-Net can no longer capture coordinated inter-finger constraints and therefore performs well only on flat, planar objects. When the Shear-Force module is ablated, the model collapses toward predicting smoothed (averaged) states; this remains workable for light objects (Lego and mouse) but fails on heavier objects because shear feedback (critical for slip detection) is absent. Without human's heuristic force from TaFo-Net, the hand-crafted strategy did not perform well on most of these tasks.

Table 3: Success rate (%) of ablated baselines across different objects.

| Method | Lego | Mouse | Book | Keyboard | Large Can | Small Can |
|---|---|---|---|---|---|---|
| Wrist-Only | 13.3 | 0.0 | 33.3 | 20.0 | 0.0 | 0.0 |
| Wrist-Only* | 0.0 | 73.3 | **100.0** | **100.0** | 6.7 | 10.0 |
| w/o Cross-Finger | 13.3 | 3.3 | 63.3 | 50.0 | 0.0 | 3.3 |
| w/o Shear-Force | **70.0** | 66.7 | 33.3 | 13.3 | 0.0 | 0.0 |
| w Heuristic Force | 36.7 | 43.3 | 66.7 | 0.0 | 0.0 | 0.0 |
| DexMove (Ours) | 66.7 | **86.7** | 90.0 | 90.0 | **63.3** | **70.0** |

## 5.3 DISCUSSIONS ON ROBUSTNESS OF DEXMOVE

To further assess the robustness of DEXMOVE, we examine its performance under more diverse conditions, including deformable objects, uneven surface and using markerless method to track objects.

**Deformable Objects.** We evaluated a rag doll and a tissue packet. Across 30 trials per object, the success rates were 96.7% and 100%, respectively. Qualitative results are shown in Fig. 5. These experiments suggest that the compliant nature of deformable objects can help stabilize contact formation and consequently achieving higher performance than previous rigid objects tested.

**Uneven Surfaces.** In everyday scenarios, support surfaces may be uneven or non-continuous. To emulate such settings, we created an uneven surface by randomly stacking additional objects beneath the manipulated items. We conducted 30 trials each with three objects: a book, a large can, and a LEGO brick. The scene and reported performances are shown in Fig. 6. We evaluate under two conditions. (1) *w/o finetune*: the original policy weights are used without any retraining. (2) *w/ finetune*: we collect 15 minutes of tactile data on

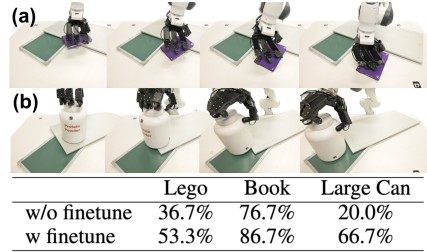

| | Lego | Book | Large Can |
|---|---|---|---|
| w/o finetune | 36.7% | 76.7% | 20.0% |
| w finetune | 53.3% | 86.7% | 66.7% |

Figure 6: Evaluation on uneven surface.

uneven surface to finetune TaFo-Net. Additionally, to simulate cases where the fingers momentarily lose contact with the object during manipulation, we mask intervals of the contact positions in the force-aware trajectory data when finetuning the DexMove policy.

Table 4: Effect of tactile noise on force prediction error (**Err**) and success rate of DexMove (**SR**).

| Noise $\sigma$ | Err-MSE (book) | SR-% (book) | Err-MSE (large can) | SR-% (large can) |
|---|---|---|---|---|
| 0 | 0.0112 | 90.0% | 0.0351 | 63.3% |
| 0.05 | 0.0108 | 86.7% | 0.0615 | 56.7% |
| 0.1 | 0.0415 | 80.0% | 0.1239 | 43.3% |
| 0.2 | 0.1721 | 53.3% | 0.3005 | 20.0% |
| 0.4 | 0.3219 | 13.3% | 0.5312 | 3.3% |

**Markerless object pose estimation.** Our approach does not necessarily require ArUco markers. To demonstrate this, we integrate FoundationPose (Wen et al., 2024) as pose estimator, under a single-camera setting with markerless objects. Qualitative results are shown in Fig. 7. Our system achieves success

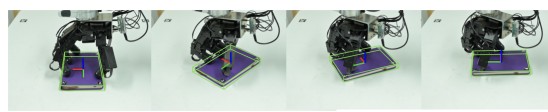

Figure 7: Visualization of markerless object pose estimation.

rates of 16.7%, 13.3%, 93.3%, 96.7%, 60.0%, and 76.6% to manipulate the six objects evaluated in Table 3, indicating that it remains effective under a single-camera and markerless setup. However, hand–object occlusions can degrade pose estimation accuracy especially for small objects. This limitation is widely recognized in the vision community and is not specific to our system.

### 5.4 Sensitivity of System To Tactile Noise

We also evaluate the influence of tactile noise and the prediction errors of predicted force, which may arise when the tactile sensor is poorly calibrated or when TaFo-Net introduces prediction inaccuracies. To emulate such cases, we add Gaussian noise $\epsilon \sim \mathcal{N}(0, \sigma^2)$ of varying magnitudes into each channel of the tactile vector field, which is normalized to $[-1, 1]$. We then measure the force prediction error of TaFo-Net and evaluate its influence on the DexMove policy by comparing the resulting task success rates. Each setting is tested over 30 trials using both the book and the large can. The results are summarized in Tab. 4. Our system maintains strong performance even when the noise standard deviation reaches 0.1, demonstrating strong tolerance to tactile noise. This robustness likely comes from two factors: the tactile signals in the training data are already noisy, enabling TaFo-Net to learn denoising, and the random dropout of time steps, fingers, and markers (in Sec. 4.3) during training further enhances the robustness of the DexMove policy.

### 5.5 Applications

The application scope of DexMove is broad. We highlight three representative scenarios: (i) Structured sorting. As shown in Fig. 5 (c), the system follows language instructions such as "move box A to region 1" and reliably transports the box to the designated zone. (ii) Language-driven human–machine collaboration. By leveraging a vision–language model (SoFar from Qi et al. (2025)), natural language commands (e.g., "put the grip of the electric drill into a person's hand") are converted into a 3-DoF target pose that serves as the goal for our policy, as shown in Fig. 5 (d). (iii) Tidying up a Desk. Given a predefined desktop layout, the system automatically relocates each item to its assigned position, as shown in Fig. 5 (e). Across these scenarios, DexMove demonstrates robust manipulation capabilities applicable to diverse real-world settings.

## 6 Conclusion and Limitations

We presented **DexMove**, a data-driven framework for dexterous non-prehensile manipulation. Our approach utilizes a hybrid data synthesis pipeline that combines the scalability of simulation for generating diverse trajectories with the realism of human demonstrations for multi-finger force control. The core of our method is the data collection methods and a set of policies that handle establishing contact, predicting future tactile force profiles with TaFo-Net, and generating goal-conditioned trajectories via DexMove-Policy. Experiments show that DexMove achieves higher success rates and efficiency than single-contact and ablated baselines, and generalizes to long-horizon, language-conditioned tasks.

Several limitations were observed in our framework. (1) Objects with articulated parts, such as a telephone with a movable handset, can shift during manipulation and destabilize contact. (2) Spherical objects tend to roll, making stable initial contact difficult and increasing the risk of slippage. (3) Certain hand poses may also cause failure, for example, pushing a tall can while grasping only its lid can cause the object to topple and restrict its rotational motion. In future work, we plan to address these issues and further explore skills that integrate both prehensile and non-prehensile techniques.

## REPRODUCIBILITY STATEMENT

We are committed to ensuring the reproducibility of our work. We plan to release all hardware designs (including the vision-based tactile sensors and the dexterous hand used in this paper), partial software implementations (including data generation scripts and inference code). Furthermore, even without access to the same hardware setup, researchers will be able to reproduce our results by leveraging our data generation pipeline to synthesize task-relevant datasets and train models that can be adapted to their own hardware settings. Our submitted supplementary video presents an extensive suite of real-robot experiments that bolsters the reproducibility and credibility of the paper's findings.

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

## A    APPENDIX

### A.1    TACTILE SENSOR

Our system employs vision-based tactile sensors on both the manipulator (fingertips of the Allegro Hand (Robotics, 2025)) and the wearable data collection device (Fig. 8). In this section, we describe the design of the proposed tactile sensors. The overall design is inspired by the open-sourced R-Tac Lin et al. (2025), which maps light intensity to physical depth. Each sensor consists of four main components: a monochrome camera module, white illuminations, a black-coated elastomer with white marker arrays, and the sensor shell.

**Camera Module.** The camera module is positioned at the base of the sensor. We use a CMOS OV9281 global shutter camera with a 160° FoV lens to capture the deformation of the curved elastomer surface under LED illumination. The camera exposure is manually fixed to ensure consistent readings. The module connects to a desktop computer via USB and outputs single-channel MJPG data at a resolution of $640 \times 480$ pixels and a frame rate of 120 fps.

**Illumination.** Illumination is provided by a white LED ring embedded in the black sensor shell. Light passes through dedicated pathways and is diffused by a frosted semi-transparent plate to achieve uniform illumination of the elastomer surface. The annular PCB hosts 8 evenly spaced 2835 4000K white surface-mounted LEDs and 470 Ω resistors, powered at 5 V.

**Coated Elastomer.** The elastomer comprises multiple silicone layers: a transparent PDMS base (Dow Corning Sylgard 184) and a semi-transparent layer (Smooth-On Ecoflex 00-10). All parts are fabricated using gel-casting techniques. The resulting curved elastomer provides a relatively uniform optical background when viewed from the camera. A black coating (Smooth-On Psycho Paint) is applied to block stray light. Compared to Lin et al. (2025), one improvement is the addition of visual marker arrays on the elastomer surface, enabling shear force detection. The white marker arrays are painted manually with a marker pen.

**Sensor Shell.** The sensor shell houses all components while forming internal light pathways. It is 3D printed in black PLA material.

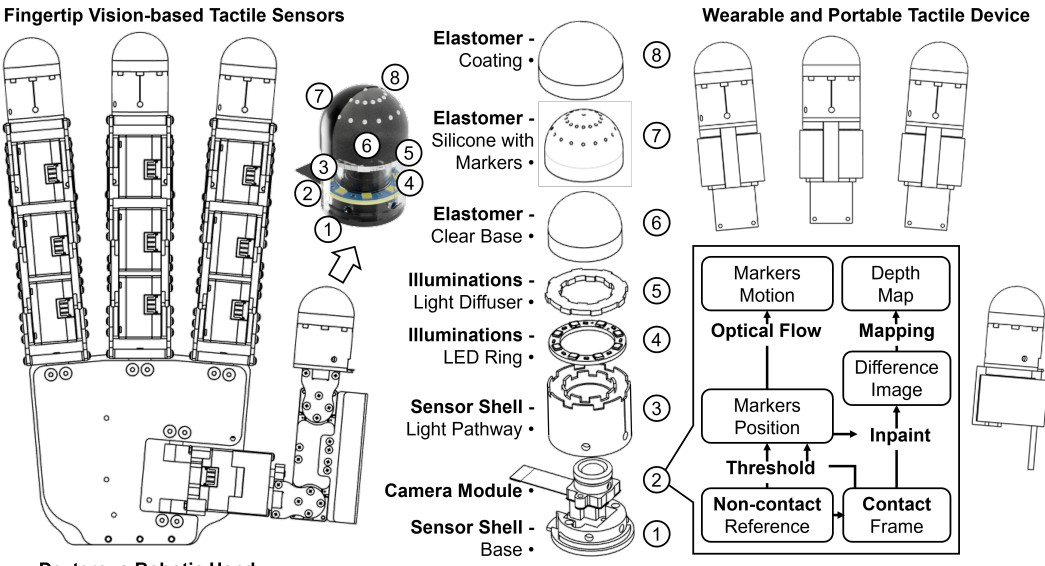

Figure 8: Monochrome vision-based tactile sensors integrated into a dexterous robotic hand and a wearable device. The sensors estimate normal forces from reconstructed depth maps and shear forces from marker displacements.

### A.2    PROCESSING TACTILE SENSOR DATA

The sensor data processing pipeline consists of two components: marker motion tracking and depth reconstruction.

**Marker Motion Tracking.** Grayscale video frames are first acquired from the camera. Adaptive thresholding is applied to segment the markers from the background. To estimate marker displacements over time, we employ the Farneback optical flow algorithm (Farnebäck, 2003), which compares the reference frame with the deformed frame to compute the flow field.

**Depth Reconstruction.** Prior to reconstruction, marker regions are inpainted in both the reference and deformed images. Then, we map grayscale values to indentation depth, a calibration procedure is performed. Following Lin et al. (2023), a 2 mm diameter spherical indenter is used to press the elastomer surface, producing a look-up table between grayscale differences and externally measured indentation depths. The final depth map is obtained by combining the relative indentation depth with the reference curved surface.

## A.3    Additional Implementation Details of System Deployment

### A.3.1    System Setup

**Camera Setup.** The flow matching policy for establishing contact is conditioned on the object point cloud. To obtain this point cloud, we use three depth cameras (Realsense D435i). The system setting is shown in Fig. 9. One camera is mounted near the elbow of the robotic arm, while the other two are positioned on opposite sides of the experimental platform to reduce occlusions and provide broader point cloud coverage. The side-mounted cameras are also used for object pose tracking. All cameras are calibrated with respect to the robot base frame. An identical configuration is applied in simulation, where virtual cameras are placed with the same extrinsic parameters to ensure geometric consistency between real and simulated environments.

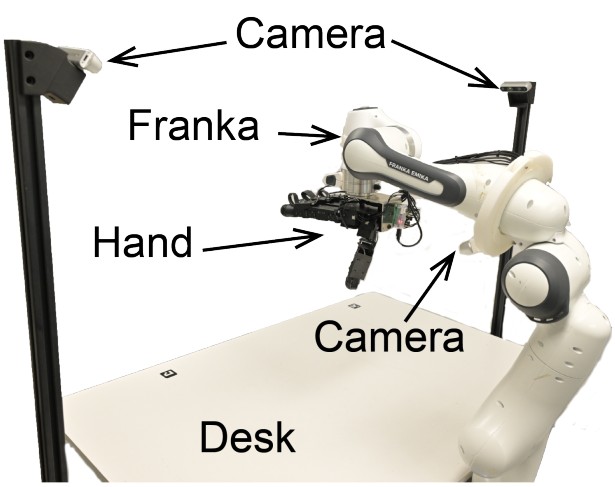

Figure 9: **System Setting:** Overview of the experimental scene, showing the relative positions of the three cameras, the desk, the Franka arm, and the hand.

**Control of the Robotic Arm and Dexterous Hand.** Both the robotic arm and the dexterous hand are controlled using position control through ROS. The robotic arm operates in Cartesian space, controlling the end-effector's position, while the hand is controlled in joint space. Joint-wise position control is implemented using PID controllers.

**Algorithm Deployment.** The policy is deployed on an NVIDIA RTX 4090 GPU, with an average inference time of approximately 22ms for each action chunk during manipulation (DexMove-Policy + TaFo-Net). At each inference step, the algorithm processes the latest $T_p = 5$ frames of sensor data to predict $T_f = 5$ consecutive actions, which are executed at 30 Hz to control both the Franka FR3 robot arm and the hand.

### A.3.2    Detail of Establish Contact

At the start of each trial, the robotic arm was moved to a collision-free configuration with an unobstructed field of view. The hand was returned to a flattened neutral "zero" pose, and the arm was positioned at a resting location to prevent fingertip occlusion of the central camera, ensuring a clear view of the workspace. We then captured synchronized depth images from three calibrated cameras and reconstructed the scene point cloud using the known intrinsics and extrinsics. A target object mask was obtained in each RGB view using SAM2 (Ravi et al., 2024), and the intersection (logical AND) of these multi-view masks was applied to the fused point cloud to remove points not belonging to the target object, effectively performing shape-from-silhouette filtering.

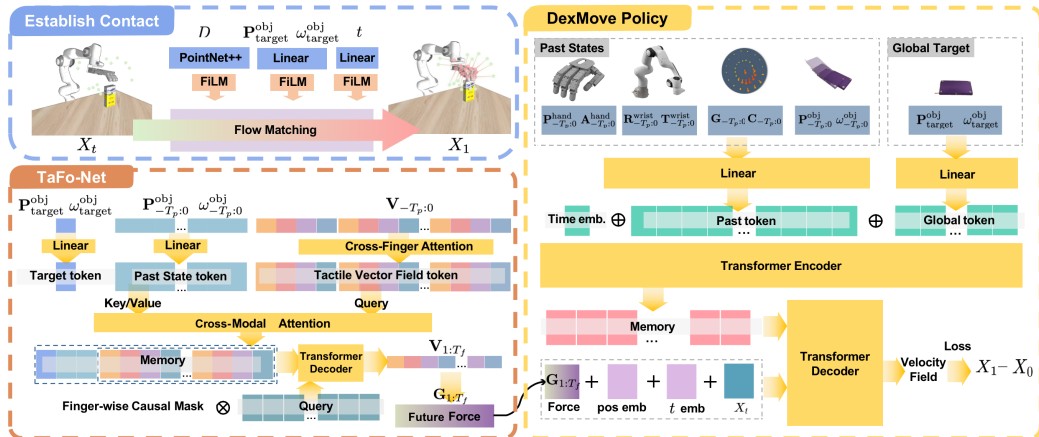

Figure 10: **Pipeline Detail:** the whole pipeline consists of establishing contact, DexMove-Policy, and TaFo-Net. We use yellow arrows to show the data flow.

This object point cloud, together with the target object pose, was used to compute a feasible contact pose. The outputs of this computation included $\mathbf{A}^{\text{hand}}, \mathbf{R}^{\text{wrist}}, \mathbf{T}^{\text{wrist}}$, where $\mathbf{A}^{\text{hand}}$ controlled the hand, while $\mathbf{R}^{\text{wrist}}$ and $\mathbf{T}^{\text{wrist}}$ specified the arm's end-effector pose in Cartesian space. In this part, all of the objects' positions and finger joints' positions are given in the arm base coordinate frame.

To minimize the risk of collision during contact establishment, the end-effector trajectory was adjusted. Instead of moving directly from the initial configuration to the contact pose, the arm was first guided to a waypoint located above the target contact pose, and then descended to the final contact configuration.

Flow matching is parameterized by a five-layer MLP with layer widths of 128, 128, 512, 1024, and 1024. The PointNet++ backbone outputs a 1024-dimensional point-cloud feature. We train with a batch size of 128 using AdamW with a learning rate of $1 \times 10^{-4}$ for 1.3M optimization steps.

### A.3.3 DETAIL OF DEXMOVE-POLICY

As shown in Fig. 10, we project the $T_p$ past frames into a sequence of past tokens, and map the global target $(P_{\text{target}}, \theta_{\text{target}})$ into a single target token (global token) using a linear layer. Concatenating these yields $(T_p + 1)$ tokens. We then append a continuous time embedding (Fourier features followed by an MLP) as an additional token, resulting in a total of $(T_p + 2)$ tokens. A Transformer encoder processes this sequence to produce a memory $\mathbf{M} \in \mathbb{R}^{(T_p+2) \times d}$.

We sample an interpolation time $t \sim \mathcal{U}(0, 1)$ and construct a noised hand state $\mathbf{X}_t = t\mathbf{X}_1 + (1 - t)\mathbf{X}_0$. We linearly project both $\mathbf{X}_t$ and the planned future target force $\mathbf{G}_{1:T_f}$ to $d$-dimensional embeddings and fuse them via FiLM (Perez et al., 2017) as query tokens. Each query token is element-wise enriched by adding a learnable positional embedding and a learnable time ($t$) embedding before being fed into the Transformer decoder, which outputs the velocity field.

During object manipulation, the two side-mounted cameras continuously tracked the object pose. Once the object reached the target pose, the inference process was terminated, and the task transitioned to the repositioning stage.

We first train the model for 200,000 iterations with a batch size of 2,048. We then perform an additional 20,000 iterations of ReFlow-based(Liu et al., 2022) fine-tuning, which compresses the sampler to a 10-step inference scheme.

### A.4 OBJECT POSE TRACKING

Since the primary focus of this work is manipulation rather than perception, the object pose is obtained using a marker-based tracking scheme. Multiple ArUco markers of known size are affixed to the object's surface, ensuring that at least one marker remains visible to the cameras at all times and enabling continuous pose estimation throughout the manipulation task. Using calibrated cameras,

we directly reconstruct each marker's world-space coordinates. Before starting, the object coordinate frame is defined by setting the origin at the centroid of all markers and aligning the orientation with the world frame. During operation, the object origin is computed from any detected marker by applying the stored marker-to-origin offset corresponding to its ID.

## A.5 IMPLEMENTATION OF BASELINES

### A.5.1 DYWA & CORN

We acknowledge the DyWA (Lyu et al., 2025) and CORN Cho et al. (2024) projects for publicly releasing their code bases, which we can run and compare. The DyWA framework first optimizes a teacher network under full privileged information (object point cloud, task state, and associated physical parameters), and subsequently distills this policy into a student restricted to obtainable robot observations. For fair comparison, we expose the object pose to the student during distillation, which simplifies and stabilizes the learning process.

CORN shares the same simulation environment as DyWA and also needs to track the object pose. To ensure a fair comparison, we further enhanced CORN by replacing its shallow MLP-based point cloud encoder with the same vision backbone as ours (PointNet++ Qi et al. (2017)). During training, we utilize 323 objects from DyWA, plus the 264 objects (88 objects already existed in YCB and were excluded) from DexMove.

### A.5.2 TELEOPERATION

In our ablation study, we conducted an experiment to control the hand wrist (with all fingers locked) via teleoperation. This was achieved using an exoskeleton (Fig. 11) with the same kinematic structure as the Franka robotic arm, scaled to half size (1:2). Each joint of the exoskeleton is equipped with a Dynamixel actuator that directly measures joint angles. These measurements are mapped to the corresponding joints of the Franka arm for execution. Experimental evaluation shows that the teleoperation system can operate at frequencies exceeding 60 Hz. For consistency with the experimental settings in this paper, we conducted teleoperation at a control frequency of 30 Hz.

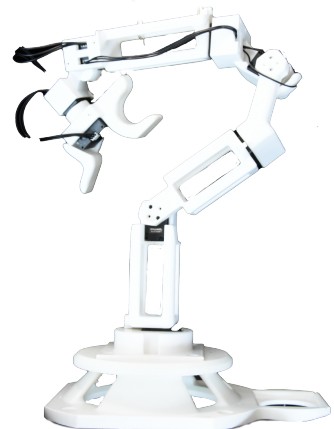

Figure 11: The exoskeleton designed at half scale (1:2) with the same kinematic structure as the Franka arm.

### A.5.3 ABLATION STUDY: WRIST-ONLY

In the ablation study, we evaluate a wrist-only controller by locking all finger joints. Concretely, during trajectory data generation, once the force-aware trajectory is obtained, we exclude the finger joint angles from the optimization variables and optimize only the wrist state ($\mathbf{R}^{\mathrm{wrist}}$, $\mathbf{T}^{\mathrm{wrist}}$). Although, for many trajectories, wrist-only motion cannot attain the optimal solution, we nevertheless use the resulting trajectories to retrain the DexMove-Policy and report the corresponding execution performance.

## A.6 OTHER SETTINGS AND RESULTS

Fig. 12 shows the six objects used in our experiments and the four tape types with distinct coefficients of friction used to construct the Friction–B tabletop. We show a more detailed execution sequence in Fig. 13 and in the supplementary video.

## A.7 DATASET

We provide additional details of the dataset through illustrative examples shown in Fig. 14 and in the supplementary video.

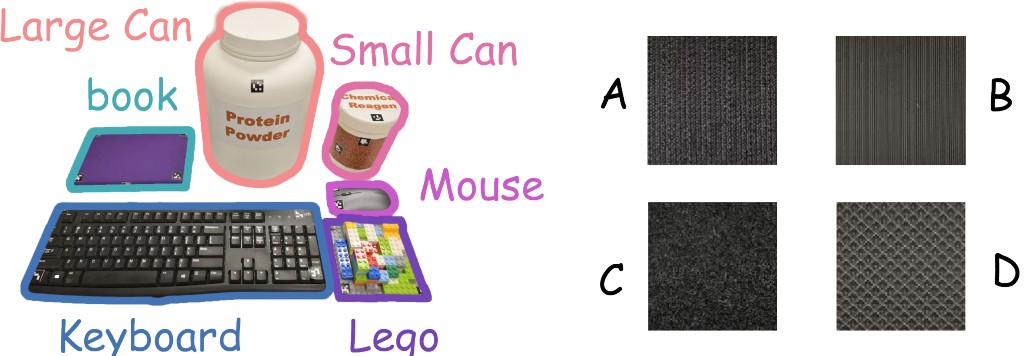

Figure 12: **Experimental Objects and Surface Friction:** The left panel shows the six objects used in the experiments, while the right panel illustrates the four types of tapes we used to construct friction B.

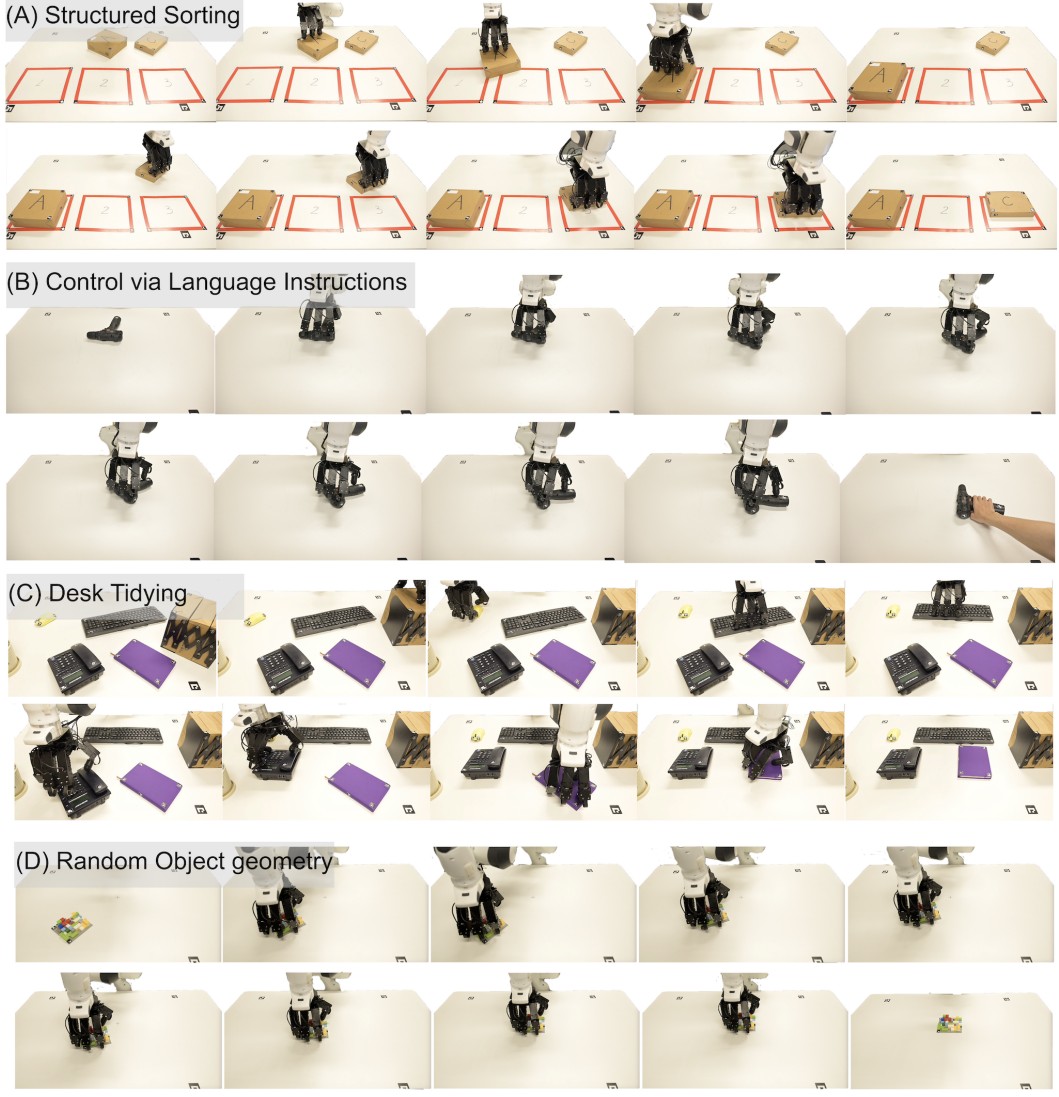

Figure 13: **Gallery of Non-Prehensile Manipulation:** Illustration of the experimental scenarios, including (a) relocating different boxes to designated regions; (b) controlling the DexMove via a large language model; (c) tidying up objects on a desk; and (d) evaluating the DexMove on objects with random geometries.

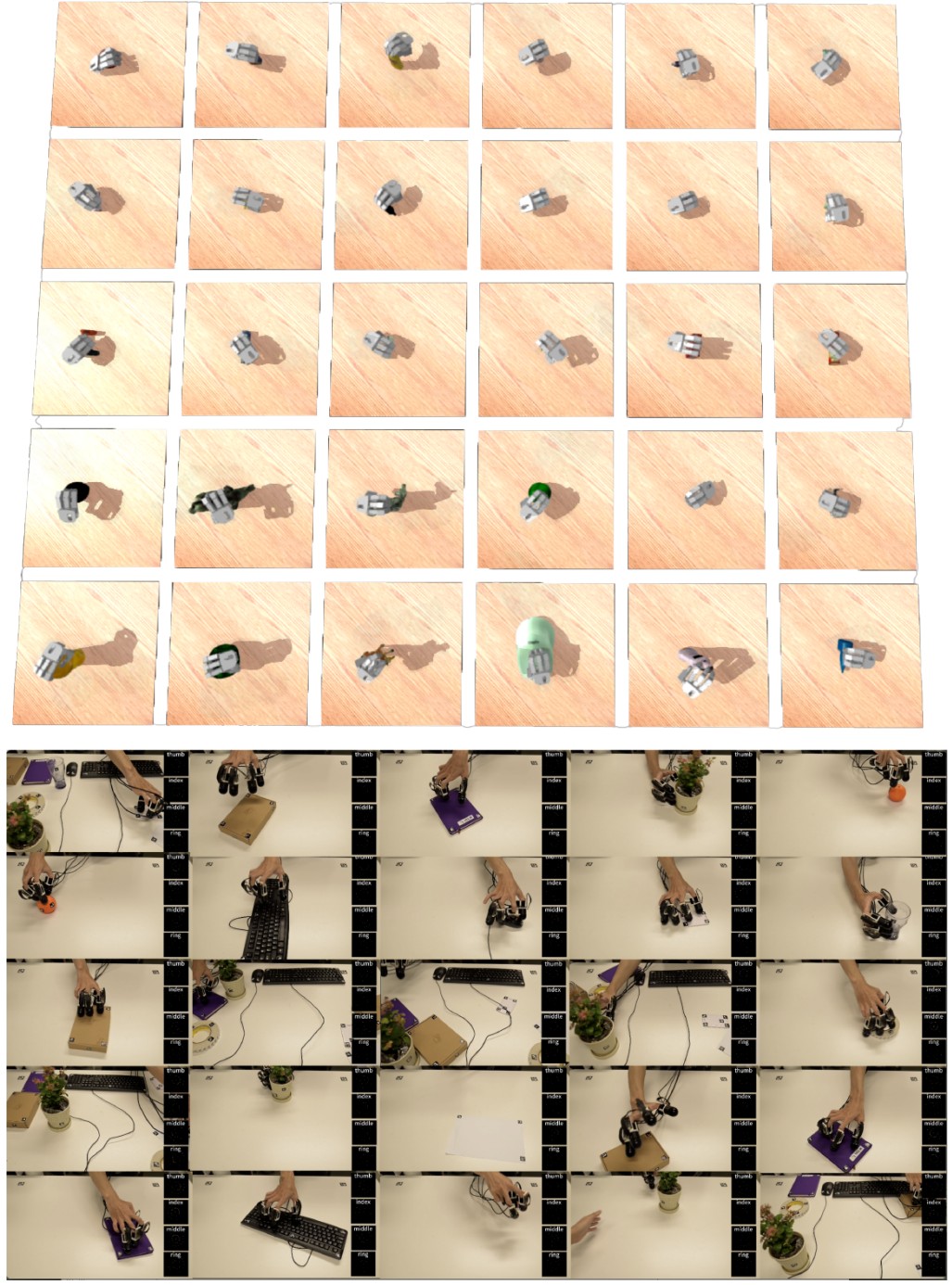

Figure 14: Illustration of the two datasets collected in this work. **Top** shows the non-prehensile trajectory dataset, **bottom** depicts the human-demonstrated tactile-force dataset.

Table 5: Summary of notations used in this paper.

| Notation | Description |
|---|---|
| $\mathbf{P}_0^{\text{tip}} \in \mathbb{R}^3$ | Initial fingertip contact position in the world frame |
| $\mathbf{P}_{0:T}^{\text{tip}} \in \mathbb{R}^{T \times 3}$ | Fingertip trajectory sequence under no-slip assumption |
| $\mathbf{P}_t^{\text{obj}} \in \mathbb{R}^3$ | Object reference point (e.g., centroid) at time $t$ |
| $\mathbf{P}_{0:T}^{\text{obj}}$ | Object translation trajectory from start to goal |
| $\omega_{\text{target}}^{\text{obj}} \in \mathbb{R}$ | Target yaw angle of the object |
| $\mathbf{R}_z(\omega) \in SO(3)$ | Rotation matrix around $z$-axis with angle $\omega$ |
| $\mathbf{R}^{\text{wrist}} \in SO(3)$ | Wrist rotation matrix |
| $\mathbf{T}^{\text{wrist}} \in \mathbb{R}^3$ | Wrist translation |
| $\mathbf{A}^{\text{hand}} \in \mathbb{R}^J$ | Joint angles of the dexterous hand with $J$ joints |
| $\mathbf{d}$ | Displacement vector from fingertip to nearest object surface point |
| $\hat{\mathbf{d}}$ | Perturbed displacement vector with Gaussian noise |
| $\mathbf{d}^{\text{tip}}$ | DIP–tip displacement vector |
| $L_{\text{region}}$ | Loss encouraging contact within sensor effective region |
| $G$ | Fingertip normal force magnitude |
| $D^{\text{sensor}}$ | Indentation depth measured by tactile sensor |
| $\vec{\mathbf{n}}$ | Surface normal at the contact point |
| $\mathbf{V} \in \mathbb{R}^{v \times 4}$ | Tactile vector field ($v = 33$ markers, encoding shear + normal) |
| $F$ | Number of fingers (4 in Allegro Hand) |
| $T_p$ | Number of history frames used as input |
| $T_f$ | Number of future frames to predict |
| $\mathbf{C} \in \mathbb{R}^{F \times 3}$ | Contact point positions in fingertip local frames |
| $\mathbf{G} \in \mathbb{R}^F$ | Normal force for each finger |
| $\mathbf{V}_{-T_p:0}$ | Historical tactile vector fields |
| $\mathbf{V}_{1:T_f}$ | Predicted future tactile vector fields |
| $\mathbf{X}_t$ | System state token at time $t$ (for flow matching) |
| $u(\cdot)$ | Velocity field learned by flow matching |
| $L_{\text{contact}}$ | Contact policy loss |
| $L_{\text{rec}}$ | Reconstruction loss of TaFo-Net |

## B  NOTATION

The summary of notions used in this paper is illustrated in Tab. 5.

## C  DECLARATION OF LLM USAGE

During the preparation of this paper, large language models (LLMs) were employed to assist with language polishing and improving the clarity of the manuscript. The models were not used for generating novel research ideas, designing experiments, analyzing results, or drawing conclusions. All scientific contributions, including the formulation of methods, implementation of algorithms, experimental design, and analysis of results, were carried out entirely by the authors.

