# OpenReview forum: "DexMove: Learning Tactile-Guided Non-Prehensile Manipulation with Dexterous Hands"
_ICLR.cc/2026/Conference — ICLR 2026 Poster_

### Official Review · Reviewer_Y6vV · 2025-10-27

**Soundness:** 3
**Presentation:** 2
**Contribution:** 3
**Rating:** 4
**Confidence:** 4

**Summary:**

The authors introduce DexMove, a framework for tactile-guided, non-prehensile dexterous manipulation. The core problem addressed is the lack of large-scale datasets and coordinated wrist-finger control policies for such tasks. The authors propose a hybrid data acquisition pipeline that combines a simulation engine for generating wrist-finger trajectories with a wearable device for capturing real-world tactile force data from human demonstrations. This data is used to train a flow-matching model to establish initial contact, a network called TaFo-Net to predict future contact forces, and the main DexMove-Policy, which generates goal-conditioned wrist-finger motions. DexMove surpasses a set of baselines across six objects and demonstrates generalization to language-conditioned long-horizon tasks in the real-world experiments.

**Strengths:**

- The hybrid data acquisition strategy is clever and effective. The hardware design is thoughtful, and the scale of the generated datasets is substantial.
- The experimental evaluation is comprehensive and rigorous. The quantitative comparison against learning-based baselines (CORN and DyWA) is strong. The supplementary video demo is very helpful and provides qualitative examples
- The demonstration of downstream applications, including structured sorting and language-driven collaboration, showcases the practical utility and downstream usages.

**Weaknesses:**

- The presentation is a bit hard to chew on. There are a few consistency and clarity issues throughout Sections 3 and 4. For example, $d^{\mathrm{tip}}$ in Eq. 2 is undefined. In Sec. 3.2, the author defines $V \in \mathbb{R}^{v \times 4}$, but in Sec. 4.3, the historical tactile data is denoted as $V_{-T_p : 0} \in \mathbb{R}^{T_p F \times vC}$. This reuse of the same symbol with different dimensions could be confusing. $P^{\mathrm{hand}}$ and $A^{\mathrm{hand}}$ seem to be redundant as one can always derive $P^{\mathrm{hand}}$ with FK.
- The overall framework is composed of three separate neural networks (Figure 4). Although it is validated through system-level evaluation, the authors do not provide a clear justification for this modular decomposition over a more integrated architecture. Moreover, the DexMove-Policy is conditioned on a planned force schedule $G_{1:T_f}$ predicted by TaFo-Net. This could lead to compounding errors, yet no evaluation of the system's sensitivity is reported. The choice of flow matching is also presented without justification over the cited diffusion policy. A brief discussion on why flow matching is preferred would be greatly appreciated.
- A disentanglement of the data contributions would further enhance the paper's quality. An experiment showing performance when training only on simulation data (e.g., with heuristic forces) is a critical missing baseline to validate this hypothesis. The claim that the isomorphic sensor design "minimizes the domain gap" is also not validated.

**Questions:**

See weaknesses

---

> ### Author Response · Authors · 2025-11-26
>
> We sincerely appreciate your valuable feedback and appreciate you recognizing the value of our work in data design and comprehensive experiment. We have made every effort to address your concerns with detailed revisions. We have addressed questions and weak points below:
>
> ---
>
> ### Weakness 1:
> Thank you for pointing out inconsistencies and unclear notation.
> - In the revised manuscript, we carefully checked all symbols and ensured that none are reused across definitions.  We have confirmed the definition of $\mathbf{V}\_{-T_p:0}$ has a typo and corrected it as $\mathbf{V} \_{-T_p:0} \in \mathbb{R}^{T \_p\times F \times v\times C}$. Please refer to our highlighted content in our revised manuscript. Other presentation issues have also been carefully reviewed and corrected.
> - $\mathbf{P}^{\text{hand}}$ and $\mathbf{A}^{\text{hand}}$: While FK and IK implicitly relate the two terms, explicitly feeding both into the network leads to a feature-redundancy representation which can rich the input space. This enables the policy network to autonomously extract the most relevant features whether positional or angular[1].
> ---
>
> ### Weakness 2:
> (1)  **Why decomposed modular design:**
>
> Our choice of a decomposed modular design, rather than an end-to-end architecture, is primarily driven by the **data limitations inherent to dexterous non-prehensile manipulation**. Fully end-to-end learning would require large-scale data that simultaneously capture hand kinematics and accurate forces, yet obtaining such data reliably remains extremely challenging. Pure simulation cannot provide realistic force signals due to the difficulty of modeling multi-contact interactions, and collecting large amounts of real-world exoskeleton data requires complex retargeting and substantial human labour.
>
> To overcome these constraints, **we adopt a hybrid strategy that lies between the two extremes.** By combining real demonstrations for learning force information with simulated trajectories for learning kinematic structure. Our decomposed design greatly reduces the required data volume and avoids the need for perfect hand–tactile alignment. More importantly, our experiments show that explicitly learning force information is beneficial and leads to strong performance across diverse tasks.
>
> Looking ahead, we are very interested in pursuing a fully end-to-end approach. With improved methods for scalable real-world data collection, such as higher-fidelity tactile sensing, vision–tactile fusion, or better hand retargeting, we believe end-to-end learning will become feasible and is a natural next step for future work.
>
> (2) **Analysis of compounding error:** We sincerely thank you for this valuable suggestion regarding evaluating how TaFo-Net’s errors influence the overall system. **We have added Table 4 in the revised manuscript**, showing that the DexMove policy remains robust under a certain range of errors in TaFo-Net’s force predictions. Further discussion of this robustness can be found in **Section 5.4**.
>
> (3) **Why flow matching policy is preferred:** We choose flow matching primarily because it achieves a significantly higher inference frame rate after reflow. In our tests, each inference step with flow matching takes only **22 ms** (in Appendix A.3.3), whereas diffusion requires **74 ms** under the same settings. Additionally, the training loss for flow matching decreases more smoothly than that of diffusion, consistent with the conclusions of [2].
>
> ---
>
> ### Weakness 3:
> (1) **A baseline trained solely on simulation data.** This baseline has already been reported in Section 5.2. Specifically, the original **“w/o Human-Guided-Force”** setting disables TaFo-Net and replaces it with a **heuristic force** strategy as same as you suggested, where the system incrementally increases applied force whenever slip is detected. We realized that the naming may have caused confusion and have updated it to **“w Heuristic Force”** in the revised manuscript for clarity.
>
> (2) **Why the isomorphic sensor design minimizes the domain gap:** In Section 3.2, we clarify that using the same sensor on both the robot and the wearable device ensures that identical external forces produce consistent sensor responses, allowing the robot to accurately learn forces from human demonstrations. Using heterogeneous sensors would make it non-trivial to map outputs between devices, introducing an additional domain gap. Recent research supports the benefits of this isomorphic sensor setup for human-to-robot skill transfer (e.g., DexOp[3]).
>
> ---
>
> We sincerely thank you again for your valuable feedback, which has helped us improve the quality of the paper.
>
> ---
>
> [1]  Redundancy-aware Action Spaces for Robot Learning, Pietro Mazzaglia, et al., RAL
>
> [2] Flow Matching for Generative Modeling, Yaron Lipman, et al.
>
> [3] DexOp: A Device for Robotic Transfer of Dexterous Human Manipulation, Hao-shu Fang, et al.

---

### Official Review · Reviewer_xjea · 2025-10-30

**Soundness:** 3
**Presentation:** 4
**Contribution:** 4
**Rating:** 8
**Confidence:** 4

**Summary:**

This paper presents DexMove, a novel data-driven framework for learning tactile-guided non-prehensile manipulation using dexterous hands. It proposes a hybrid data collection pipeline that combines large-scale simulation of wrist–finger trajectories with real-world human demonstrations collected via a wearable tactile device. Based on the data collection pipeline, it further introduces a three-part policy structure that is able to establish contact, predict future tactile force, and generate tactile-informed trajectories to handle a set of non-prehensile manipulation tasks. Experiments show that DexMove achieves higher performance than existing baselines, and generalizes to long-horizon, language-conditioned tasks.

**Strengths:**

This paper is well motivated. The proposed hybrid data pipeline that fuses large-scale simulation with human tactile demonstrations shows potential to provide high-fidelity datasets for the community. Real-world experiment results demonstrate DexMove's superiority over single-contact and ablated baselines, showing its generation ability towards generalizing to long-horizon, language-conditioned tasks.

**Weaknesses:**

1. The proposed method relies on strong assumptions: it employs three calibrated depth cameras and visual markers to track object pose and obtain visual observations.

2. The paper lacks a detailed discussion of common failure cases. Understanding how and why the method fails (e.g., specific object geometries, friction conditions, or loss of contact) is critical for the audience.

3. The demonstration of language-conditioned tasks like "tidying" is presented as a qualitative strength, but it lacks quantitative evaluation. It is unclear how robust this capability is, what the success rate is for these more complex tasks, and how much the language instruction actually guides a complex rearrangement versus simply triggering a pre-learned pushing skill.

**Questions:**

Please refer to the Weaknesses part.

---

> ### Author Response · Authors · 2025-11-26
>
> We faithfully appreciate your positive rating and further comments that help us improve paper quality. We have seriously considered all weakness and improve our manuscript as follows:
>
> ---
>
> ### W1: Calibrated depth cameras and visual markers
> Thank you for raising this important point. Our deployment is not restricted to a three depth-camera setup or the use of visual markers. **In the revised manuscript (Sec. 5.3), we added a new experiment demonstrating deployment under a markerless scenario, where object pose is estimated using FoundationPose [1] with a single view.** The DexMove policy continues to perform successfully under this setup. However, we note that multiple cameras can provide better robustness against hand–object occlusions, especially when manipulating small objects. In such cases, a single camera may occasionally lose track of the object pose.
>
> ---
>
> ### W2. Failure case analysis
>
> **In the revised manuscript (Sec. 6),** we provide a discussion of the most common failure modes. We observed that DexMove tends to fail in two scenarios: **(1)** when manipulating objects with **unstable structures**, such as articulated objects whose components may separate during motion, and **(2)** when handling **spherical objects** that easily slip out of the hand due to limited contact stability.
>
> We also observed that DexMove achieves less than 70% success on the Large Can. The main reason is that manipulating this object often requires two incompatible contact poses. To rotate the can, the hand may ideally contact the lid; whereas to translate it stably, the hand need to grasp the body. When both rotation and translation are required in the same sequence, it becomes intractable to maintain a single contact pose that supports both motions, which leads to failures
>
> ---
>
> ### W3. Quantitative evaluation on language-conditioned tasks
> We sincerely thank you for pointing this out. We have included quantitative evaluations of our framework using a large language model in the task "tidying".
> We benchmarked "tidying" over 10 trials per scene, with each scene two to five objects were randomly shuffled. After marking both the current and target poses of these objects in the image, we feed this image to GPT-5 and ask GPT-5 to output the execution order of these obects. Our policy was then used as the actor to execute the planned sequences. The manipulation and planning success rates are summarized below:
>
> | **Number of objects to manipulate** | **2**    | **3**    | **4**   | **5**   |
> | ------------------------------------- | ---------- | ---------- | --------- | --------- |
> | **Planning Success Rate**           | 100% | 100% | 80%| 80% |
> | **Manipulation Success Rate**       | 90%  | 80% | 70% | 60% |
>
> An incorrect plan may cause the manipulated object to collide with previously placed objects, leading to task failure. Errors in the pushing order inferred by the LLM are reflected in the **Planning Success Rate**, whereas the **Manipulation Success Rate** measures whether all objects are ultimately repositioned successfully.
>
> ---
>
> We sincerely thank you again for your time and effort in helping us improve the manuscript. Please feel free to let us know in case any points that are unclear.
>
> ---
> [1] FoundationPose: Unified 6D Pose Estimation and Tracking of Novel Objects, Bowen Wen, Wei Yang, Jan Kautz, Stan Birchfield, CVPR 2024

---

### Official Review · Reviewer_byvP · 2025-10-31

**Soundness:** 3
**Presentation:** 2
**Contribution:** 3
**Rating:** 6
**Confidence:** 4

**Summary:**

This paper presents a tactile-guided framework for non-prehensile manipulation with dexterous hands. The authors propose a complete pipeline that combines large-scale simulation-generated wrist–finger trajectories with human tactile demos to learn stable contact and motion strategies. The system integrates three core components: ContactFlow Matching for establishing robust contact using flow matching; DexMove-Policy, a goal- and force-conditioned flow-based policy that generates motion trajectories; and TaFo-Net, a tactile force planner modeling inter-finger coordination over time. DexMove enables the robot to manipulate various objects through pushing, rolling, and sliding without grasping, achieving high success rates and efficiency in real-world tests.

**Strengths:**

1. The paper tackles the underexplored problem of tactile-guided non-prehensile manipulation using dexterous hands, a setting rarely addressed in prior work that mostly focuses on grasping or gripper-based pushing.  The combination of large-scale simulation-based wrist–finger trajectory generation with human tactile demonstrations is both conceptually novel and practically valuable, enabling scalable yet physically grounded learning.
2. The technical design is solid and coherent: three modules are integrated through a consistent flow-matching formulation, supported by clear ablations and real-world evaluations.
3. The paper presents its ideas with clear structure and illustrative figures, making complex mechanisms understandable.

**Weaknesses:**

1. Limited Evaluation Scope and Generalization Evidence – Although real-world experiments demonstrate promising success rates, the object set remains small and relatively homogeneous (rigid, medium-sized items). The absence of tests on soft, slippery, or deformable objects limits claims of generalization.
2. The experimental evaluation mainly covers lateral pushing, sliding, and simple rotational tasks, but lacks richer contact interaction types such as pressing, squeezing, or compliant surface exploration, which are essential to demonstrate the full potential of tactile-guided dexterous control. Expanding the benchmark to include pressing or deformable-object manipulation would provide stronger evidence of versatility.
3. Several parts of the paper, especially the method section, are densely written without clear intuitive explanation. The logical flow among the three modules (ContactFlow, DexMove-Policy, and TaFo-Net) could be clearer. A cleaner, layered schematic emphasizing data flow and module dependency would make the overall system architecture much easier to understand. Some parts of the article could be polished.

**Questions:**

1. How sensitive is the learned policy to tactile sensor calibration or noise? Since the method heavily relies on tactile feedback, it would be useful to know how performance changes under slight calibration errors or sensor drift over time Some sensitivity analysis is recommended.
2. Could the proposed approach be extended to handle dynamic or moving objects? I'm just curious about whether DexMove can adapt its tactile-guided policy to scenarios where the object itself is in motion (e.g., sliding or being perturbed), and what modifications would be required to handle such cases.
3. Plz see weaknesses.

---

> ### Author Response · Authors · 2025-11-26
>
> We sincerely thank you for recognizing the strengths of our work, including the **comprehensive system design** and our **novel data pipeline**. We respond to each question below, followed by detailed revisions made to address the paper’s weaknesses.
>
> ---
>
> ### Q1. System sensitivity of tactile noise
> We sincerely thank you for this valuable suggestion regarding evaluating how sensitive our system is to tactile noise. **We added specific experiments and discussion in  Section 5.4 of our revised manuscript**. Specifically, we added Gaussian noise of varying scales into tactile vector field to mimic the noisy input may caused by calibration error and calculate its influence on the predicted force of TaFo-Net and the success rate of the whole system. **Please refer to Table 4 in our revised manuscript.**
>
> ---
>
> ### Q2. Extended to handle dynamic or moving objects
> Following your suggestion,  we introduced a new setting where **manual disturbances (pushing and blocking)** are applied during manipulation of six different objects. We conducted 30 trials for each object, and the quantitative results are summarized below:
> |                  | **LEGO**  | **Mouse** | **Book**  | **Keyboard** | **Large Can** | **Small Can** |
> | ------------------ | ----------- | ----------- | ----------- | -------------- | --------------- | --------------- |
> | **Success rate** | 60.0% | 66.7%| 93.3% | 93.3%    | 50.0%     | 56.7% |
>
>
> In our experiments, we observed that when pushing or blocking caused the object to slip out of the hand, DexMove could not complete the task because it currently lacks a re-grasping mechanism. However, when the object remained within finger contact, DexMove successfully completed the manipulation, benefiting from its closed-loop control design.
>
> ---
>
> ### W1 & W2.  Broader Evaluation Scope
> We agree that a broader evaluation scope would better demonstrate the potential of our system and have implemented the experiments in the revised manuscript. **Following your suggestion, the revised manuscript now includes experiments with soft and deformable objects (e.g., a rag doll and a tissue pack) and objects manipulated on uneven surfaces in Section 5.3**. We thank you for bringing attention to this important evaluation concern. In addition, we would like to highlight that the objects used in our previous experiments were not homogeneous in scale. The smallest object (mouse) measures only 8 cm × 5 cm × 5 cm, while the largest (Large Can) measures 20 cm × 20 cm × 28 cm.
>
> ---
>
> ### W3. unclear representation
> Thank you for the valuable suggestion. We  have corrected some unclear expressions and will further polish the paper to make it clearer and easier to understand. We refer to our updates in Sec. 3 and Sec. 4 of our revised manuscript.
>
> ---
>
> We sincerely thank you for your time and positive assessment of our work. If you have any further questions, please feel free to let us know.

---

### Official Review · Reviewer_9vkj · 2025-11-09

**Soundness:** 3
**Presentation:** 3
**Contribution:** 3
**Rating:** 6
**Confidence:** 4

**Summary:**

This paper proposes DexMove, a tactile-guided non-prehensile manipulation framework for dexterous hands. It first generates force-aware wrist-finger trajectories in simulation, and then collects real tactile data via a glove-like exoskeleton system, where a human wears the same tactile fingertips as the robot. Using the simulated trajectories, the authors train a force-conditioned imitation learning policy. Then, using real-world trajectories, the system learns to predict the desired forces given a target object pose and uses these predictions to guide the imitation policy. The method achieves robust planar pushing across diverse tabletop objects and generalizes to higher-level, language-conditioned tasks such as sorting and tidying. Experiments show a 77.8% success rate, outperforming gripper-based baselines and ablations by a large margin.

**Strengths:**

Comprehensive system design: The paper demonstrates an impressive integration of simulation, tactile sensing, human demonstration, and policy learning. The wearable tactile exoskeleton represents substantial engineering effort.

Novel hybrid data pipeline: The combination of simulated trajectories and tactile demonstrations effectively mitigates the lack of large-scale real-world tactile data and the sim-to-real gap.

Compelling demonstrations: Real-world experiments, including language-guided and long-horizon tasks, highlight its potential as a general dexterous manipulation system.

Connection to VLMs: Integration with vision-language models for goal specification expands the scope of tactile manipulation toward
multimodal reasoning and autonomy.

**Weaknesses:**

Task simplicity: Despite the complex hand control, the main tasks remain planar pushing on a tabletop. The method’s advantage over simpler mechanisms (like parallel-jaw pushers) is somewhat limited by the task design.

Pose tracking with markers: Object pose estimation relies on markers, reducing realism and preventing deployment in unstructured environments.

Limited wrist motion: The dataset penalizes wrist rotations beyond ~90°, which may restrict generalization to more complex manipulations.

Evaluation diversity: The test objects and surfaces, while varied, do not yet cover dynamic or multi-object interactions; thus, generalization remains partially demonstrated.

System scalability: The wearable system and tactile glove require manual calibration and mounting. The scalability and robustness of data collection remain uncertain.

**Questions:**

1. How many different policies are trained in total? Specifically, are the contact-establishment and DexMove-Policy networks trained separately or jointly?

2. Are the contact and trajectory policies both trained entirely in simulation?

3. How robust is perception to occlusion, given the reliance on three depth cameras? Does failure of one camera degrade policy performance?

4. In Table 3, what exactly does “Wrist-only*” denote? Does that mean teleoperation-based wrist control with locked fingers (as opposed to a policy-based wrist-only controller)?

5. Have the authors tested beyond planar pushing, such as rolling, sliding on uneven surfaces, or multi-object rearrangement?

6. More importantly, comments on the sim-to-real gap: Is TaFo-Net essential for bridging this gap? If so, why? Is it because force information is critical, or simply because real-world tactile data are needed to fine-tune the network?

---

> ### Author Response · Authors · 2025-11-26
> **Response to Reviewer 9vkj (1/2)**
>
> We sincerely thank you for your thoughtful feedback and for recognizing the strengths of our work, including the **hybrid data pipeline**, the **comprehensive system design**, and the **compelling demonstrations**. Below we address the specific questions and clarify how we have further strengthened the work in response to your concerns.
>
> ---
>
> **Q1:** We train three policies: (1) Establish Contact, (2) TaFo-Net, and (3) DexMove-Policy. For all presented tasks, the same trained weights are used without any task-specific fine-tuning. The Establish Contact policy and DexMove policy are trained separately
>
> ---
>
> **Q2:** Yes. Contact establishment and kinematic trajectories are fully synthesized in simulation, and both the Establish Contact policy and the DexMove policy are trained using these simulated data. Human demonstrations are used only to learn real-world force interactions by TaFo-Net.
>
> ---
>
> **Q3:** We agree that occlusion affects robustness, particularly during the contact establishment stage, which relies on point clouds. Our use of multiple cameras is specifically intended to mitigate this issue. The other stages remain robust to partial occlusion (including hand–object occlusion) because they only require pose tracking via ArUco. In practice, we found that three cameras provide sufficient redundancy, and occlusion did not hinder execution. However, when using a single camera with FoundationPose without ArUco, the system becomes less robust, as shown in our new robustness evaluation experiment of the revised manuscript.
>
> ---
>
> **Q4:** Yes. In this baseline, the wrist is teleoperated while all fingers remain fixed. Its degraded performance demonstrates that non prehensile manipulation requires coordinated control of both the wrist and the fingers, and relying on wrist motion alone is insufficient to accomplish the task.
>
> ---
> **Q5 & W1 & W4:** Regarding raised concerns about limited task diversity and evaluation breadth. We expanded our experiments to include **uneven surfaces**, **multi-object rearrangement**, **deformable objects**, and **manual disturbances**:
>
> **(1) Uneven surfaces:** Objects were successfully slid on uneven surfaces, using stacked objects as support. **Results and discussion are updated in Section 5.3 of the revised manuscript.**
>
> **(2) Multi-object rearrangement** is demonstrated in our supplementary **video** and discussed in the **Section 5.5**.
>
> **(3) Deformable objects:** a rag doll and a packet of tissues were tested; 30 trials each yielded success rates of 96.7% and 100%, respectively. **Qualitative results are updated in Section 5.3 of the revised manuscript.**
>
> **(4) Manual disturbances:** We applied manual perturbations (pushing and blocking) during manipulation of six objects. For each object, 30 trials were conducted. DexMove demonstrates robustness against such disturbances due to its closed-loop policy design. Failures occurred only when objects slipped completely out of the hand; this could be addressed in future work by incorporating a re-grasping policy.
>
> |                  | **LEGO**  | **Mouse** | **Book**  | **Keyboard** | **Large Can** | **Small Can** |
> | ------------------ | ----------- | ----------- | ----------- | -------------- | --------------- | --------------- |
> | **Success rate** | 60.0% | 66.7% | 93.3% | 93.3%   | 50.0%    | 56.7%     |
>
> ---
>
> **Q6:** Yes, TaFo-Net is crucial for bridging the sim to real gap. This is because accurately obtaining the desired contact forces that serve as the reference for motion control is highly challenging but crucial. Incorrect force predictions can cause the object to slip out of the hand or lead to undesirable force competition between fingers, ultimately destabilizing the manipulation.
>
> For **model based strategies such as heuristic forces**, it is difficult to reliably predict how a change in the pressure of one finger influences the forces on other fingers. As shown in Table 3, removing TaFo-Net and replacing it with heuristic forces (originally “w/o Human Guided Force” and now referred to as “w Heuristic Force”) leads to a drastic performance drop and makes the task nearly infeasible.
>
> **Using simulation** to obtain ground truth forces is also unreliable. Multi contact interactions are difficult to model due to unknown object inertia, inaccurate friction parameters, and limitations of numerical solvers. As a result, force information learned purely from simulation does not transfer well to real hardware.
>
> **TaFo-Net addresses this challenge by learning real world force information from human demonstrations. This real force prior effectively bridges the remaining gap and significantly improves control performance.**

---

> > ### Author Response · Authors · 2025-11-26
> > **Response to Reviewer 9vkj (2/2)**
> >
> > **W2: Using markers to track.**
> > We agree that marker-less perception would simplify deployment. To this end, we added a new experiment using a single RGB camera with FoundationPose [1], demonstrating that a marker-free setup is feasible. **The updated results are included in the Section5.3 of the revised manuscript.**
> >
> > ---
> >
> > **W3:** This constraint prevents workspace violations and encourages the controller to rely more on finger forces. For tasks requiring larger wrist rotations, the constraint can be disabled. Alternatively, objects can be manipulated in multiple steps, analogous to human strategies, as human wrists are similarly constrained.
> >
> > ---
> >
> > **W5:**  We agree that wearable demonstrations require human involvement; however, in our hybrid pipeline, they constitute only a **small** fraction of the training data. Simply **two hours** of human demonstrations are sufficient to achieve the reported success rates. Moreover, collecting human demonstrations is often more efficient than teleoperation data, making the approach scalable. For tactile sensor calibration, the procedure in [2] can be followed easily. The mechanical design of the wearable system (see appendix) allows straightforward mounting and deployment. No ergonomic issues were observed during use.
> >
> > ---
> >
> > We sincerely thank you again for their comments, which have helped us improve the quality of the manuscript.
> >
> > ---
> >
> > [1] FoundationPose: Unified 6D Pose Estimation and Tracking of Novel Objects, Bowen Wen, Wei Yang, Jan Kautz, Stan Birchfield, CVPR 2024
> > [2] PP-Tac: Paper Picking Using Tactile Feedback in Dexterous Robotic Hands, Pei Lin, Yuzhe Huang, Wanlin Li, Jianpeng Ma, Chenxi Xiao, Ziyuan Jiao, RSS 2025

---

### Author Response · Authors · 2025-12-04
**Message to  Area Chair and all Reviewers**

We sincerely thank the reviewers for their thoughtful and constructive feedback. We are grateful that the reviewers found our work **well-motivated and addressing an under-explored yet important challenge** (R2, R3), that **our data-acquisition strategy is novel and effective**(R1, R2, R3, R4), that **our technical design is solid and coherent** (R1, R2), that **our experimental evaluation is comprehensive** (R1, R3, R4), and that **our downstream applications are practical** (R1, R3, R4). During the rebuttal process, we have carefully addressed all reviewer concerns, including new evaluations and typo corrections as reflected in the updated manuscript. Specifically:
- To strengthen experimental evaluation, we added **4** additional experiments that quantitatively and qualitatively demonstrate the robustness of DexMove. These include tests on **deformable objects, non-flat surface conditions, a single-camera setting and human disturbance**. These experiments directly address concerns from R1 and R2 regarding the diversity and coverage of DexMove’s evaluation.
- To address questions about the stability of our modular design (R2, R4), we added **new experiments analyzing how errors from different modules, particularly tactile noise, propagate through the system**. These experiments systematically quantify the impact of increasing tactile noise and prediction error on overall control performance, and the results show that DexMove remains robust even under substantial levels of tactile noise.
- To enhance clarity, we **clarified all of the presentations** in Section 3 and Section 4 (R4), and improved the narrative motivation and design insights (R2).
- To address concerns about the limits of our approach (R3), we have **added a detailed analysis of failure cases**, specifically investigating the causes of lower success rates for certain objects during real-world experiments.

We believe these revisions significantly improve the quality of the manuscript. We sincerely thank the Area Chair and reviewers for their time and constructive feedback.

---

### Meta-Review · Area_Chair_L6fy · 2026-01-18

**Summary:**

The main concerns across reviewers centered on whether the proposed system demonstrates sufficient task diversity and generalization beyond planar non-prehensile manipulation, the strength of the perception assumptions (multi-camera setup and markers), the justification and robustness of the modular architecture, and clarity issues in the technical presentation. Some reviewers also questioned whether the demonstrated tasks fully justify the use of a dexterous hand over simpler end-effectors, and whether the language-conditioned experiments were sufficiently evaluated quantitatively.

**Reviewer Concerns:**

Several major concerns were convincingly addressed in the rebuttal and revised manuscript. The authors substantially expanded the experimental evaluation, adding tests on deformable objects, uneven surfaces, multi-object rearrangement, robustness to tactile noise, manual disturbances, and a single-camera, marker-less perception setup. These additions directly respond to concerns about generalization, robustness, and sensitivity to noise and occlusion. The authors also provided new analyses quantifying how errors in tactile force prediction affect downstream control, alleviating concerns about compounding errors in the modular design. Presentation and notation issues raised by multiple reviewers were corrected, and a clearer failure case analysis was added. In addition, the previously qualitative language-conditioned demonstrations were complemented with quantitative results.

Some concerns remain only partially resolved. Despite the broader evaluation, the core manipulation behaviors are still dominated by planar pushing, sliding, and rotation, and richer dexterous interactions are not extensively explored. Questions about the long-term scalability and deployment burden of the wearable tactile system remain largely qualitative. Finally, while new baselines and ablations strengthen the case, some reviewers’ skepticism about whether a dexterous hand is strictly necessary for the demonstrated tasks is only partially addressed.

**Reviewer Scores:**

- Reviewer 9vkj would likely raise their score slightly, as most technical questions and evaluation gaps were directly addressed with new experiments and clearer justification.
- Reviewer byvP would likely remain around the original score or increase marginally, given that evaluation scope and robustness concerns were addressed, while task richness concerns persist.
- Reviewer xjea would likely maintain their positive score, as their main weaknesses were addressed in the revision.
- Reviewer Y6vV would likely increase their score modestly, since presentation issues, missing baselines, and robustness analyses were explicitly addressed, though some architectural concerns may remain.

---

### Decision · Program_Chairs · 2026-01-26

Accept (Poster)